# Molecular insights into the origin of the Hox-TALE patterning system

Bruno Hudry[1]*, Morgane Thomas-Chollier[2], Yael Volovik[3], Marilyne Duffraisse[4], Amélie Dard[4], Dale Frank[3], Ulrich Technau[5], Samir Merabet[4]*

[1]MRC Clinical Sciences Centre, Faculty of Medicine, Imperial College London, London, United Kingdom; [2]Unité Mixte de Recherche (UMR) 8197, INSERM U1024, Institut de Biologie de l'ENS (IBENS), Ecole Normale Supérieure (ENS), Centre National de Recherche Scientifique (CNRS), Institut National de la Santé et de la Recherche Scientifique (INSERM), Paris, France; [3]Department of Biochemistry, The Rappaport Family Institute for Research in the Medical Sciences, Faculty of Medicine, Technion, Israel Institute of Technology, Haifa, Israel; [4]Unité Mixte de Recherche (UMR) 5242, Institut de Génomique Fonctionnelle de Lyon (IGFL), Ecole Normale Supérieure (ENS) de Lyon, Centre National de Recherche Scientifique (CNRS), Lyon, France; [5]Department für Molekulare Evolution und Entwicklung, University of Vienna, Vienna, Austria

*For correspondence:
petithudry@gmail.com (BH);
samir.merabet@ens-lyon.fr (SM)

Competing interests: The authors declare that no competing interests exist.

**Abstract** Despite tremendous body form diversity in nature, bilaterian animals share common sets of developmental genes that display conserved expression patterns in the embryo. Among them are the Hox genes, which define different identities along the anterior–posterior axis. Hox proteins exert their function by interaction with TALE transcription factors. Hox and TALE members are also present in some but not all non-bilaterian phyla, raising the question of how Hox–TALE interactions evolved to provide positional information. By using proteins from unicellular and multicellular lineages, we showed that these networks emerged from an ancestral generic motif present in Hox and other related protein families. Interestingly, Hox-TALE networks experienced additional and extensive molecular innovations that were likely crucial for differentiating Hox functions along body plans. Together our results highlight how homeobox gene families evolved during eukaryote evolution to eventually constitute a major patterning system in Eumetazoans.

## Introduction

'What is an animal?' In 1993, Slack et al. proposed to define an animal by the zootype (*Slack et al., 1993*), a concept illustrating the strong conservation of embryonic expression profiles of developmental genes observed in different bilaterian phyla at that time. Since then, it was found that developmental genes could also display highly dissimilar expression patterns or even be absent in non-bilaterian lineages, showing that the genetic mechanisms underlying embryonic development are not universal.

One major class of developmental genes that was historically considered as highly conserved in the animal kingdom is the Hox genes. Hox genes are expressed along the anteroposterior axis of all bilaterian animals, providing positional information during embryogenesis. Given their important patterning roles, Hox genes are thought to have strongly contributed to morphological diversification of bilaterian organisms during evolution. Accordingly, numerous examples have shown that modifications in Hox genes number, expression, and/or activity could correlate to morphological variations across bilaterian lineages (*Heffer and Pick, 2013*).

The role of Hox genes suffers more ambiguity outside Bilateria, and in particular in the sister Cnidaria group. Cnidarians do contain Hox genes and have undergone a wide range of morphological

**eLife digest** Any animal with a body that is symmetric about an imaginary line that runs from its head to its tail is known as a bilaterian. Humans and most animals are bilateral, whereas jellyfish and starfish are not. Bilateral symmetry can take many forms—as demonstrated by the differences between flies, frogs and humans—but all bilaterians express many of the same genes during development.

One of these groups of genes is known as the Hox family. The expression of specific Hox genes at specific times instructs cells in the developing embryo to adopt different fates according to their position along the anterior–posterior (head to tail) axis. The patterning function of Hox genes relies on the presence of two additional cofactors that belong to the so-called TALE family. Although both Hox and TALE proteins were present early on during animal evolution, it is unclear how and when the interactions between them first began to generate symmetrical body plans.

Now, Hudry et al. have provided insights into the origin of the Hox-TALE network by analysing the expression and molecular properties of Hox and TALE proteins from various multicellular and unicellular organisms. These experiments revealed that Hox and TALE proteins of the sea anemone *Nematostella*, which belongs to a group of animals called cnidarians that have radial rather than bilateral symmetry, interact with one another in a similar manner to the interactions seen in bilaterians.

Hudry et al. then showed that two *Nematostella* Hox genes were able to substitute for their bilaterian equivalents in fruit flies, and that a *Nematostella* TALE gene was able to take over neuronal functions of its equivalent in *Xenopus* frogs. This striking conservation of function between species suggests that Hox and TALE genes were already working together in the common ancestor of all bilaterian and cnidarian animals.

By contrast, TALE members from a unicellular amoeba were unable to interact with Hox proteins, suggesting that Hox–TALE interactions first emerged in multicellular animals. In addition to increasing our knowledge of highly conserved Hox signalling, these data provide insight into the molecular mechanisms that gave rise to the symmetrical body plan that has been adopted, and adapted, by the majority of animals since.

radiation during evolution. However, Hox genes have neither a clear collinear nor conserved expression profile along the primary (oral–aboral) axis of the planula larva among different cnidarian lineages (*Finnerty et al., 2004*; *Kamm et al., 2006*), which renders their orthology status difficult to assign (specially for posterior/non-anterior cnidarian Hox genes). A role of Hox genes in axis patterning was proposed in the hydrozoan *Eleutheria dichotoma* (*Jakob and Schierwater, 2007*), but these results are limited by the fact that functional analyses were performed in medusa, a particular developmental stage that is not shared by all cnidarian species. The role of cnidarian Hox genes during early larval stages is, however, currently unclear.

Interestingly, Cnidaria is the only non-bilaterian phylum, which has a bona fide Hox repertoire, whereas others, including ctenophores, sponges, and placozoans, lack Hox genes (*Figure 1*). This raises the important question of how the Hox gene family acquired its crucial axial patterning functions during metazoan evolution. In Bilateria, Hox patterning functions rely on the presence of the PBC and Meis proteins, which are also present in non-bilaterian phyla (*Figure 1*). We thus assessed whether a Hox/PBC/Meis network could exist outside Bilateria, and if so whether or not it would rely on identical molecular rules as observed in Bilateria.

Hox and PBC/Meis proteins belong to the ANTP (Antennapedia) and TALE (Three amino acids loop extension) class of homeodomain (HD)-containing TFs, respectively (*Saina et al., 2009*). The Hox/PBC/Meis network relies on interactions between PBC and Meis proteins on one side, and on interactions between Hox and PBC proteins on the other side (*Mann et al., 2009*). Some posterior vertebrate Hox members do form dimeric complexes with Meis (*Shen et al., 1997*; *Williams et al., 2005*), but these interactions constitute a vertebrate innovation rather than a general rule in Bilateria. Interactions between PBC and Meis occur through conserved regions localized upstream of the HD of both proteins (called PBC-A and MEIS-A domains) and which are thought to derive from a common ancestor domain (*Burglin, 1998*). These interactions allow the nuclear translocation and stability of

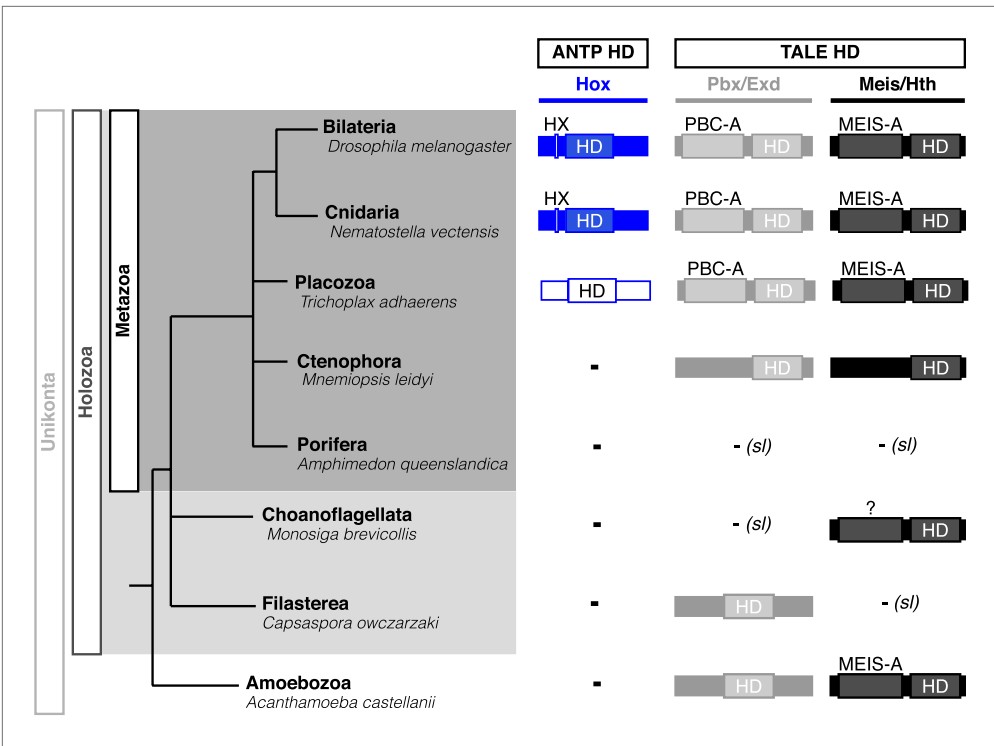

**Figure 1**. Phylogeny of Hox (ANTP superclass) and PBC/Meis (TALE superclass) proteins across eukaryote evolution. Protein motifs required for Hox/PBC/Meis network are indicated when present: homeodomain (HD), hexapeptide (HX), PBC-A, and MEIS-A. Absence of the member in a given group is considered as resulting from a secondary loss (sl), when the ortholog is present more ancestrally. A question mark is indicated for Meis of *Monosiga brevicollis* because of incomplete sequence. The protein indicated in Placozoa is not coloured in blue since it is not a true Hox protein (see main text for details). Examples are provided for a representative species of each group. Members of PBC and Meis classes are called Pbx or Extradenticle (Exd) and Meis or Homothorax (Hth), respectively.

PBC (*Abu-Shaar et al., 1999*). Interactions between Hox and PBC involve a short conserved motif in Hox proteins, called the hexapeptide (HX), which folds within the hydrophobic pocket formed in part by the extra-three residues of the HD of PBC (*Merabet et al., 2009*). This motif contains a conserved core sequence of four residues in all but AbdB-group Hox proteins, which retain a single tryptophan residue. Additional specific signatures can also be found in the HX of some paralog groups (*Merabet et al., 2009*). Recent data showed that Hox proteins could also interact with PBC partners through other more specific motifs. These alternative interaction modes can be induced by the DNA-binding of the Meis partner, eventually leading to different three-dimensional conformations that could be important for paralog-specific functions (*Galant et al., 2002*; *Merabet et al., 2007*; *Hudry et al., 2012*).

PBC and Meis representatives are found from unicellular Amoebozoa and Filasterea groups to metazoan lineages including Ctenophora and Placozoa (*Figure 1*). Sequence analysis shows that only one PBC or Meis representative is present, or that the protein does not contain the PBC-A or MEIS-A interaction domain in most of these lineages (*Figure 1*, *Figure 2—figure supplements 1 and 2*). Interestingly, all protein features required for PBC/Meis partnership appeared concomitantly with the presence of Hox or Hox-like proteins in Metazoa (*Figure 1*). Of note, the representative species of Placozoa, *Trichoplax adhaerens* (*Ta*), contains a protein that was classified as a ProtoHox (*Schierwater and Kuhn, 1998*) or ParaHox (*Mendivil Ramos et al., 2012*) member (*Figure 1*). However, the absence of any HX motif (*Schierwater and Kuhn, 1998*) suggests that this protein could not interact with PBC/Meis, which is confirmed later (see last section of 'Results').

Protein sequence analysis indicates that a Hox/PBC/Meis network could first be present in Cnidaria. To test this hypothesis, we dissected the molecular properties underlying the formation of

the Hox/PBC/Meis interaction network of the sea anemone *Nematostella vectensis* (*Nv*), a cnidarian species exhibiting an internal symmetry organized along oral–aboral (primary) and directive (secondary) body axes (*Figure 2A*). This analysis was performed in vitro and in vivo and completed by heterologous functional assays in vertebrate and invertebrate species. Since other members of the ANTP superclass are described to interact with PBC in Bilateria, we also searched for the molecular mechanisms that allowed the emergence of the Hox–TALE network during evolution of homeobox gene families.

Our results show that Hox and TALE proteins from *Nematostella* form interaction networks and perform similar functions to their bilaterian counterparts. Although these networks rely on intricate molecular properties, they originated from an ancestral generic mode of interaction that was kept in other homebox gene families. Overall our study describes how the molecular cues underlying the Hox–TALE patterning system in Bilateria was established stepwise during eukaryote evolution.

## Results

### Hox and TALE members are co-expressed and form protein complexes in the *Nematostella* embryo

The *Nematostella* genome contains seven Hox genes and one representative of the PBC (*NvPbx*) or Meis (*NvMeis*) class (*Figure 2A*). NvPbx and NvMeis proteins show a high level of sequence conservation with their bilaterian counterparts, especially in the regions encompassing the HD, PBC-A, and Meis-A domains (*Figure 2B*, *Figure 2—figure supplements 1 and 2*). In contrast, sequence similarity between cnidarian Hox proteins and their bilaterian homologs is restricted to the region encompassing the HD, as exemplified for NvHoxB (*Figure 2B*, *Figure 2—figure supplement 3*) and NvHoxE (*Figure 2B*, *Figure 2—figure supplement 4*). Some NvHox proteins do also contain a HX motif, as noticed in NvHoxB or NvHoxE (*Figure 2B*, *Figure 2—figure supplements 3 and 4*). The HX of NvHoxE is more divergent, corresponding to a single tryptophan residue as found in bilaterian Hox posterior paralog groups. Still, the identity of NvHoxE (as well as those of NvHoxF) remains controversial, being classified as a cnidarian-specific (*Chourrout et al., 2006*; *Ryan et al., 2006*), posterior (*Gauchat et al., 2000*; *Ryan et al., 2007*), or central Hox gene (*Thomas-Chollier et al., 2010*). This is not the case for NvHoxC, NvHoxDa, NvHoxDb, NvHoxA and NvHoxB, which are unambiguously assigned as anterior Hox genes (*Chourrout et al., 2006*; *Ryan et al., 2006, 2007*; *Thomas-Chollier et al., 2010*; *Figure 2A*).

To first verify that Hox, PBC, and Meis products from *Nematostella* could form an interaction network in vivo, we performed in situ hybridization experiments using *NvPbx* (*Matus et al., 2006*) and *NvMeis* probes. Results showed that the transcripts of these two genes are co-expressed in the entire endoderm of the larva (*Figure 2C–D*). Interestingly, several Hox genes were previously described to be expressed in staggered domains along the directive axis in the same tissue (*Finnerty et al., 2004*; *Matus et al., 2006*; *Ryan et al., 2007*), which was here confirmed for NvHoxB and NvHoxE (*Figure 2C–D*).

The relationship between NvPbx and NvMeis was then analysed by expressing a tagged version of NvPbx, alone or with NvMeis. Analyses were performed in a 24-hr-old embryo, at a stage when endogenous *NvMeis* is not yet expressed. We observed that the nuclear accumulation of mCherry-NvPbx fusion protein is contingent upon co-injection with NvMeis (*Figure 3A*). Thus, NvMeis is able to stabilize NvPbx, eventually triggering its nuclear accumulation. This observation is reminiscent of Pbx/Meis relationships in bilaterians (*Abu-Shaar et al., 1999*; *Saleh et al., 2000*). We further confirmed this result by visualizing NvPbx/NvMeis complexes directly in live *Nematostella* embryos through BiFC (Bimolecular Complementation Fluorescence). BiFC relies on the property of fluorescent proteins to be reconstituted when their two non-fluorescent sub-fragments are close enough in space. This method has been developed in several animal model systems to validate interactions between two candidate partners in vivo (*Kodama and Hu, 2012*). In this study, we co-expressed NvPbx and NvMeis fused respectively to the N-terminal (VN) or C-terminal (VC) fragment of Venus. This resulted in fluorescent signals in the cytoplasm (where interaction occurs first) and nuclei of embryonic cells (*Figure 3B*). No BiFC was obtained between a fusion construct and the complementary isolated VC or VN fragment, highlighting that the interaction between NvPbx and NvMeis fusion proteins was not artificially induced by the inherent affinity of the VN and VC fragments (*Figure 3B*).

BiFC was also used to visualize interactions between NvPbx and NvHoxE (*Figure 3C*). The specificity of this interaction was validated by the absence of BiFC between NvHoxE and a DNA-binding

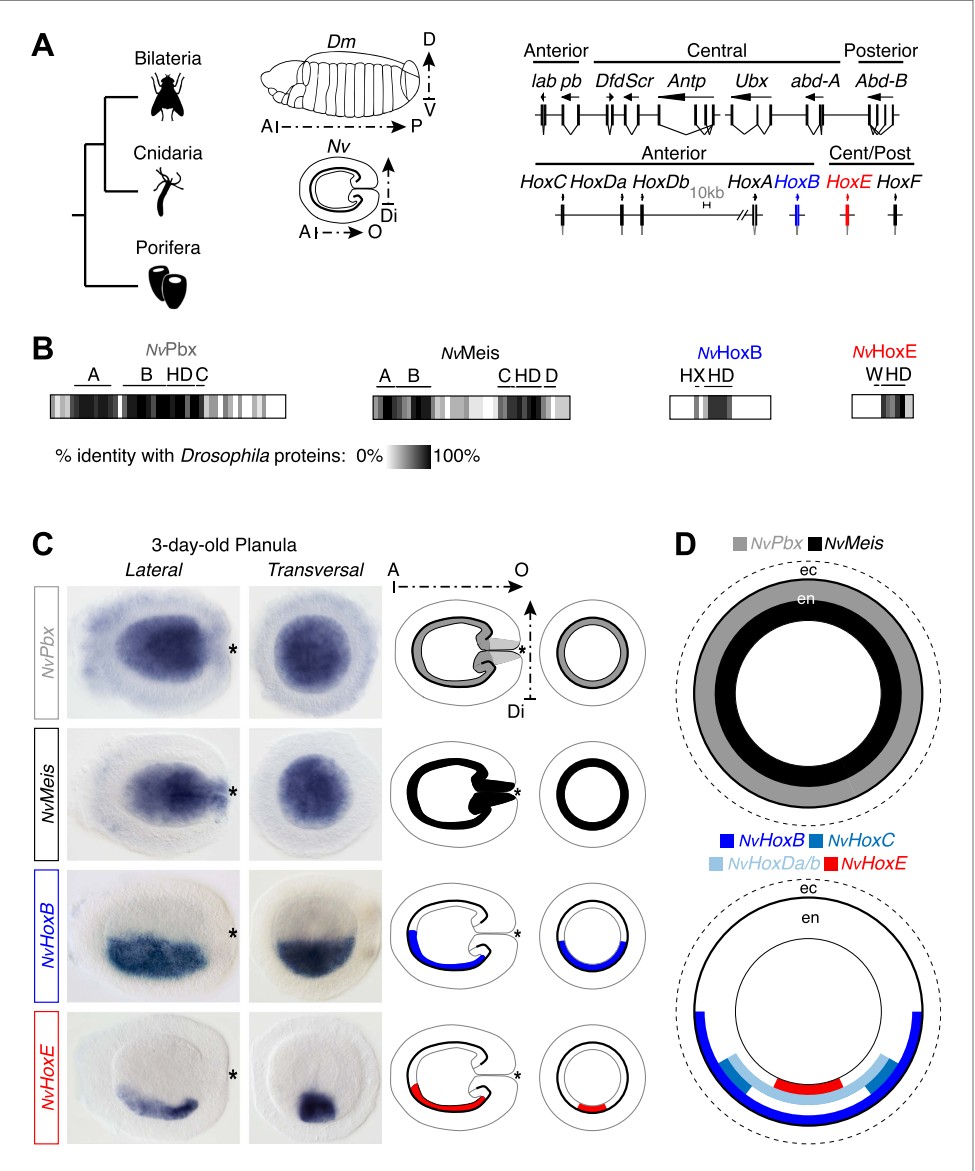

**Figure 2**. Hox and TALE members are co-expressed in the endoderm of the Nematostella embryo. (**A**) Genomic organisation of Hox genes in *Nematostella vectensis* (*Nv*) and *Drosophila melanogaster* (*Dm*), two representative species of Cnidaria and Bilateria, respectively. Embryos at the planula stage are schematized; A–P: anterior–posterior; D–V: dorsal–ventral; A–O: aboral–oral; Di: directive axis. The Nematostella embryo is oriented according to recent findings (***Sinigaglia et al., 2013***). The nomenclature is calqued on (***Chourrout et al., 2006***) to avoid confusions with bilaterian Hox paralogs: *NvHoxC* (*antHox7*), *NvHoxDa* (*antHox8*), *NvHoxDb* (*antHox8a*), *NvHoxA* (*antHox6*), *NvHoxB* (*antHox6a*), *NvHoxE* (*antHox1a*), *NvHoxF* (*antHox1*). The two Nematostella Hox genes under study, *NvHoxB* and *NvHoxE*, are highlighted in blue and red respectively. Note that the same colour code is used in other figures. (**B**) Sequence identity between Nematostella and Drosophila proteins. *Nv*HoxB and *Nv*HoxE are compared to Labial (Lab) and Ultrabithorax (Ubx) respectively. The percentage of identity is represented by a grayscale gradient. Conserved domains in bilaterian TALE proteins are indicated (**A**, **B**, **C**, **D**). HX: hexapeptide. HD: homeodomain. See also ***Figure 2—figure supplements 1–4***. (**C**) In situ hybridization of *NvPbx*, *NvMeis*, *NvHoxB* and *NvHoxE* in a three-day-old Nematostella planula. These four genes are expressed in the endoderm (en), as illustrated in (**D**). Ec: ectoderm. *NvPbx* and *NvMeis* are illustrated in grey and black, respectively. This colour code is used in other figures.

The following figure supplements are available for figure 2:

**Figure supplement 1**. Protein sequence alignment of PBC members from representative species of Unikonta.

*Figure 2. Continued on next page*

*Figure 2. Continued*

**Figure supplement 2**. Protein sequence alignment of Meis members from representative species of Unikonta.

**Figure supplement 3**. Protein sequence alignment between NvHoxB and the Labial (Lab) protein from *Drosophila melanogaster*.

**Figure supplement 4**. Protein sequence alignment between NvHoxE and the Ultrabithorax (Ubx) or AbdominalB (AbdB) protein from *Drosophila melanogaster*.

deficient form of *Nv*Pbx (**Figure 3C**), showing that the formation of the cnidarian Hox/Pbx complex is DNA-binding dependent, as previously noticed in bilaterians (**Hudry et al., 2011**, **2012**). Altogether, these results show that *Nv*Pbx and *Nv*Meis are co-expressed with several *Nv*Hox genes in the endoderm and that *Nv*Hox and *Nv*TALE proteins can constitute an interaction network in vivo.

## Interaction properties between *Nematostella* and bilaterian Hox/TALE complexes are highly similar in vitro

Next, we analysed the molecular properties underlying the assembly of *Nematostella* Hox/Pbx/Meis complexes in vitro. *Nv*Pbx and *Nv*Meis proteins were previously shown to interact with bilaterian Hox proteins (**Hudry et al., 2012**). Here, we conducted binding assays with *Nv*HoxB and *Nv*HoxE proteins and assembly properties of protein complexes were measured by electromobility shift assays (EMSAs) on three different DNA probes. DNA probes differ by one nucleotide in the Hox/Pbx binding site (**Figure 4A**), each one corresponding to the preferential DNA-binding sites of previously defined anterior, central, and posterior Hox/Pbx complexes with vertebrate and invertebrate proteins (**Shen et al., 1996**; **Slattery et al., 2011**).

We observed that *Nv*HoxB/*Nv*Pbx and *Nv*HoxE/*Nv*Pbx complexes display anterior and central DNA-binding preferences, respectively (**Figure 4B–B'**). The addition of a consensus Meis binding site in a topology found in known Hox target enhancers (**Mann et al., 2009**) confirmed that *Nv*HoxB and *Nv*HoxE do form trimeric complexes with the TALE partners on DNA (**Figure 4C**). Interestingly, the DNA-binding of *Nv*Meis is also sufficient for rescuing the loss of *Nv*Hox/*Nv*Pbx complex formation upon the HX mutation (**Figure 4C**), suggesting that *Nv*Meis is able to remodel *Nv*Hox–*Nv*Pbx interactions. Given the sequence divergence between the two *Nv*Hox proteins (**Figure 4—figure supplement 1**), these alternative interaction modes are presumably paralog-specific, as previously suggested in bilaterians (**Merabet et al., 2007**; **Hudry et al., 2012**).

## Genomic binding sites for *Nv*Hox and *Nv*TALE proteins are preferentially localized in the promoter region of genes expressed in the endoderm and allow the assembly of Hox/TALE complexes in vitro

Our band shift assays were performed on consensus binding sites previously defined with bilaterian Hox and TALE proteins. To know whether such sites could be used in the context of *Nematostella* development, we searched for their presence in the *Nematostella* genome. We predicted that these DNA-binding sites should be found in the promoter region of genes expressed in the endoderm, where Hox and TALE products are present together. By comparison, the promoter region of genes expressed in the ectoderm should not be enriched in Pbx/Meis-binding sites since the TALE partners are absent in this tissue. Therefore, we performed an in silico analysis based on 76 genes displaying a characterized developmental expression pattern during *Nematostella* embryogenesis. The choice of working with a limited number of genes was motivated by the fact that we did not find any significant enrichment of Hox/PBC/Meis binding sites in a genome-wide search, neither in *Nematostella* nor in other metazoan species (**Figure 5—figure supplement 1**). In addition, by limiting the search in non-coding regions (of at least 60 base pairs) conserved in 12 *Drosophila* genomes, we could find a higher density of Hox/PBC/Meis clusters (**Figure 5—figure supplement 1**). These results highlight that the signal to noise ratio is too low when considering Hox/PBC/Meis binding sites in all non-coding regions, and that the space search needs to be restricted to observe a significant enrichment.

Our in silico analysis revealed a significant enrichment of Hox/Pbx binding sites in the promoter region of genes expressed in the endoderm compared to the ectoderm (**Figure 5A,B**, 'Materials and methods'). Several of these genes also contain a consensus Meis binding sequence within the

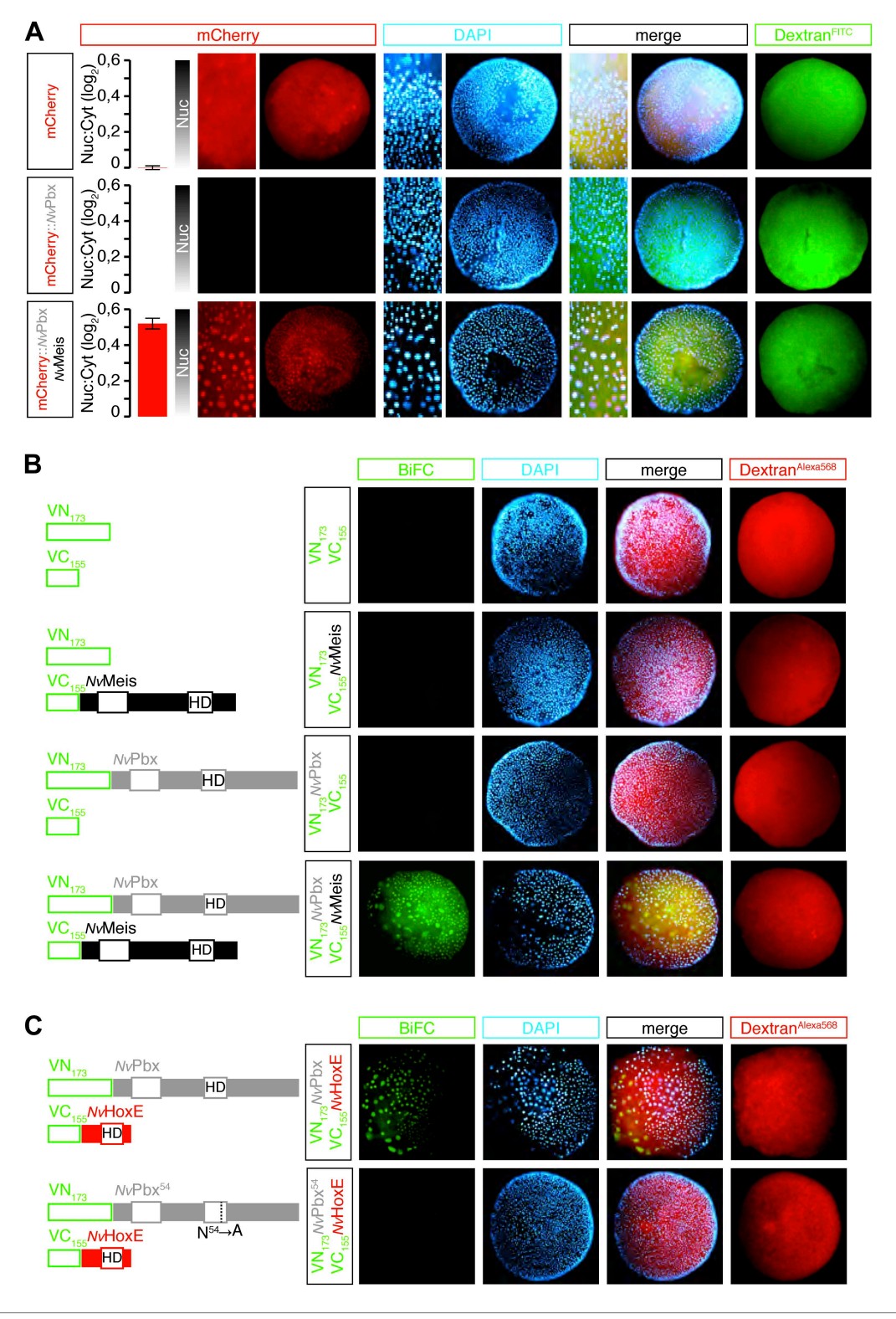

**Figure 3**. Hox and TALE members form protein complexes in vivo. (**A**) *Nv*Pbx interacts with *Nv*Meis in vivo. The nuclear localisation of a fusion mCherry-*Nv*Pbx protein was only observed upon co-injection with *Nv*Meis. Graphs on the left are quantifications of the ratio between nuclear and cytoplasmic fluorescent signals (log2). Note that mCherry-*Nv*Pbx alone did not lead to any signal, suggesting that the fusion protein is not stable in the absence of

*Figure 3. Continued on next page*

*Figure 3. Continued*

Meis, as noticed in bilaterians. (**B**) BiFC between *Nv*Pbx and *Nv*Meis in the *Nematostella* embryo. Fusion constructs are schematized on the left. VN: N-terminal fragment of Venus; VC: C-terminal fragment of Venus. Specificity of BiFC is verified by the absence of fluorescent signals upon the injection of isolated VN or VC fragments, together or with the complementary VC-*Nv*Meis or VN-*Nv*Pbx fusion proteins, as indicated. Interaction between *Nv*Pbx and *Nv*Meis occurs both in the cytoplasm and nucleus (see text for details). (**C**) BiFC between *Nv*HoxE and *Nv*Pbx in the *Nematostella* embryo. Interaction occurs only in the nucleus. Mutation of the residue 54 in the homeodomain (HD) of *Nv*Pbx abolishes DNA-binding and BiFC with *Nv*HoxE. In all panels, Dapi (cyan) stains nuclei and Dextran (red) is a control of injection conditions.

---

40 nucleotides surrounding the Hox/Pbx binding site (*Figure 5C*, 'Materials and methods'). Among the candidate target genes containing a putative Hox/Pbx/Meis binding site in their promoter region, we found *NvHoxB* and *NvHoxC*. Thus, auto/cross-regulatory loops may occur between *Nv*Hox genes, as observed in bilaterians. Since sequences surrounding Hox/TALE binding sites can also strongly influence the protein complex formation (*Ebner et al., 2005*; *Hudry et al., 2012*), we verified that the binding sites found in proximity of *NvHoxB* and *NvHoxC* could indeed allow the assembly of Hox/TALE complexes in vitro. We confirmed that *Nv*HoxB and *Nv*HoxE could form dimeric or trimeric complexes on these putative binding sites. Interestingly, *Nv*HoxB and *Nv*HoxE displayed distinct DNA-binding preferences on these two sites (*Figure 5D–D'*). Again, the assembly of *Nematostella* Hox/Pbx complexes was HX-dependent on both probes, except in the presence of *Nv*Meis, highlighting that alternative interaction modes between *Nematostella* Hox and TALE proteins can occur on various DNA-binding sites (*Figure 5—figure supplement 2*). Altogether these results show that the molecular properties underlying the Hox-TALE system are conserved between Cnidaria and Bilateria.

## *Nematostella* Hox and TALE proteins can execute generic functions in invertebrate and vertebrate species

To assess whether the *Nematostella* Hox/TALE system could have any conserved biological function, we examined the activity of *Nv*Hox and *Nv*TALE proteins in two different bilaterian organisms, the fly *Drosophila melanogaster* and the frog *Xenopus laevis*.

For Hox assays, series of EMSAs previously confirmed that *Nv*HoxB and *Nv*HoxE are able to associate with the *Drosophila* Pbx (Extradenticle, Exd) and Meis (Homothorax, Hth) cofactors on different *Drosophila* Hox target enhancers in vitro (*Figure 6—figure supplement 1*). Of note, *Nv*HoxB and *Nv*HoxE again display anterior or central-like DNA-binding preferences on those physiological target sites, respectively. Furthermore, BiFC validated that *Nv*Hox proteins could interact with Exd in vivo (*Figure 6—figure supplement 1*). Again, the specificity of BiFC was confirmed with a DNA-binding deficient form of Exd (*Figure 6—figure supplement 1*).

The activity of *Nv*Hox proteins in *Drosophila* was then measured in two generic Hox assays: the antenna-to-leg transformation in adult (*Casares et al., 1996*; *Yao et al., 1999*), and the rescue of the Hox *labial* (*lab*) mutant phenotype in a particular structure of the central nervous system called the tritocerebrum (*Hirth et al., 1998*). We found that *Nv*HoxB and *Nv*HoxE were able to successfully function as their *Drosophila* homologs in both assays (*Figure 6A,B*). The antenna-to-leg transformation by *Nv*Hox proteins was shown to rely on the assembly of a repressive trimeric complex with *Drosophila* TALE cofactors on cis-regulatory sequences of the *spineless* (*ss*) target gene (*Figure 6A*, *Figure 6—figure supplement 2*; *Duncan et al., 2010*). Moreover, *Nv*HoxB and *Nv*HoxE behave like central and anterior paralogs in the tritocerebrum, respectively (*Figure 6B*, *Figure 6—figure supplement 2*; *Hirth et al., 1998*). Finally, the activity of both *Nv*Hox proteins in the antenna and tritocerebrum appears to be dependent on the integrity of the HX motif (*Figure 6—figure supplement 2*).

The functional conservation of *Nv*Pbx was addressed by analysing its potential to rescue zygotic *exd* mutant phenotypes in *Drosophila*. As for *Nv*Hox proteins, we previously verified that *Nv*Pbx is able to form a protein complex with *Drosophila* Hox and Meis proteins in vitro and in vivo (*Figure 6—figure supplement 3*). We observed that *Nv*Pbx could rescue the *exd* mutant phenotype in the *Drosophila* larva cuticle (*Figure 6C*). Providing *Nv*Pbx in this mutant background was also sufficient to rescue the A1 transforming activity of Ubx (*Figure 6C'*), which is also known to depend on the integrity of the HX (*Galant et al., 2002*). These results highlight that *Dm*Exd and *Nv*Pbx are functionally equivalent, at least for specification functions in the *Drosophila* epidermis.

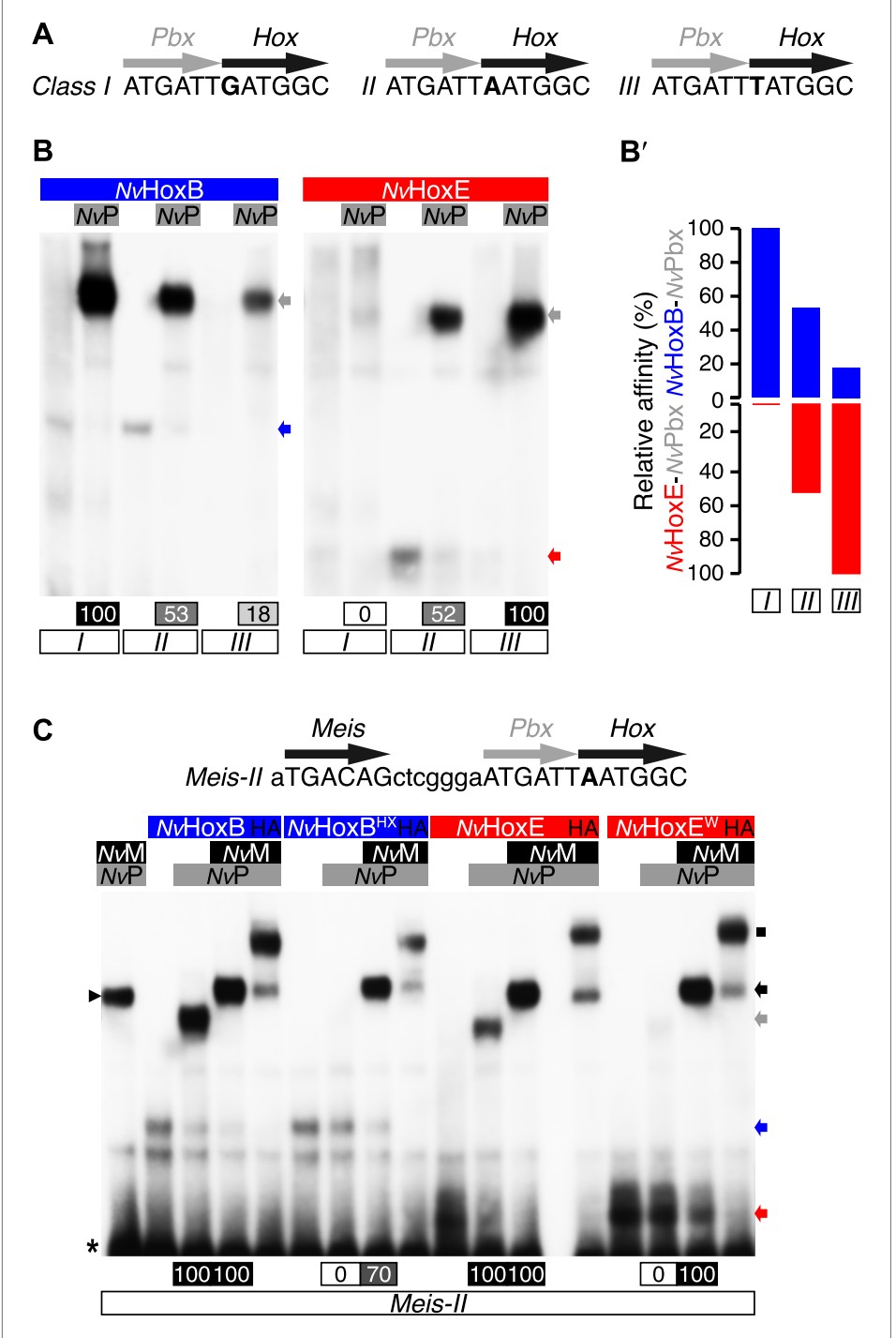

**Figure 4**. Interaction properties between *Nv*Hox and *Nv*TALE proteins in vitro. (**A**) Nucleotide sequence of the different classes of Hox/Pbx binding sites used in band shift experiments. The nucleotide that distinguishes each Hox/Pbx binding site is bolded. (**B–B'**) Band shift experiments between *Nv*HoxB or *Nv*HoxE and *Nv*Pbx on the three different classes of binding sites, as indicated. Coloured and grey arrows point to monomer or dimer binding, respectively. Graph on the right (**B'**) depicts the relative affinity of each dimeric complex on the three different binding sites, as deduced from the direct quantification on the gel (values are indicated at the bottom). (**C**) Band shift experiments between wild-type or HX-mutated *Nv*Hox proteins and *Nv*TALE cofactors, as indicated. Colour codes and annotations are as in (**B**). Black arrow indicates trimeric *Nv*Hox/*Nv*Pbx/*Nv*Meis complexes. Other bands are not specific (proteins of the lysate). Black scare highlights the supershift band resulting from the addition of an

*Figure 4. Continued on next page*

*Figure 4. Continued*

antibody against the HA tag of *Nv*Hox proteins. Asterisk shows the free probe. Note that the loss of dimeric *Nv*Hox/*Nv*Pbx complex upon the HX mutation is rescued in the presence of *Nv*Meis.
The following figure supplements are available for figure 4:

**Figure supplement 1**. Protein sequence alignment between *Nv*HoxE (upper sequence) and *Nv*HoxB (lower sequence).

Finally, the functional conservation of *Nv*Meis was measured in the *Xenopus* embryo, which constitutes a well-established developmental model system for assessing Meis activities. In particular, *X*Meis proteins are known to be required for the specification of posterior cell fates along the AP axis of the central nervous system (*Dibner et al., 2001*), a function that also involves Pbx1 (*Maeda et al., 2001*). Accordingly, ectopic expression of *X*Meis proteins causes anterior neural truncations with a concomitant expansion of hindbrain and spinal cord (*Salzberg et al., 1999*). This phenotype is reproduced with the fly or mouse Meis proteins, demonstrating that it can constitute a generic assay for assessing Meis function. We observed that the injection of *Nv*Meis in animal caps of *Xenopus* embryos was also able to robustly induce the expression of several posterior spinal cord marker genes (like *Hoxa7*, *cdx1* or *cdx2*), although to a lesser extent than *X*Meis3 (*Figure 6—figure supplement 4*). Thus, *Nv*Meis displays a striking functional similarity with its bilaterian homologs.

In sum, our assays highlight a striking functional conservation between *Nematostella* and bilaterian Hox and TALE proteins, suggesting that the Hox/TALE network was already at work in the common ancestor of Cnidaria and Bilateria.

## Genesis of Hox–TALE interaction networks across Metazoa

Among the ANTP class, we can distinguish three main subclasses that originally derived from a ProtoANTP ancestor (*Saina et al., 2009*): Hox/ParaHox, NK and extended-Hox (*Figure 7*). Consistent with their close evolutionary relationships, Hox, NK and extended-Hox members share common protein features including the presence of an HX motif upstream of the HD (*Figure 7*). These observations raise the question of the molecular mechanisms that led to the emergence of the Hox-TALE network and more generally to interaction networks between TALE proteins and other members of the ANTP class across metazoan evolution. We postulated that the HX motif could have constituted a major protein scaffold that provided an ancestrally conserved TALE interaction potential to different ANTP members.

To test this hypothesis, we started by analysing the interaction properties between NK and TALE proteins. Previous works showed that some vertebrate NK members could interact with PBC (*Brendolan et al., 2005*) or PBC/Meis (*Rhee et al., 2004*), but the role of the HX was not addressed in these interactions. Here, we analysed the molecular properties underlying complex assembly between the *Nematostella* NK representative *Nv*Msx and the two *Nv*TALE partners. We observed that *Nv*Msx is not able to interact with *Nv*Pbx except in the presence of *Nv*Meis (*Figure 8A*). Trimeric complex formation is however not as strong as with *Nv*Hox proteins and is also fully dependent on the integrity of the HX motif (*Figure 8A*).

Our results suggest that protein region(s) in *Nv*Msx could mask the interaction with Pbx in absence of Meis. We confirmed this hypothesis by testing a series of truncated and chimeric proteins generated from *Nv*Msx and *Nv*HoxB. We found that deleting the N- and C-terminal parts of *Nv*Msx allowed dimeric complex formation with *Nv*Pbx (*Figure 8B,C*). Conversely, the N- and C-terminal regions of *Nv*Msx are sufficient to alleviate the interaction between a minimal *Nv*HoxB protein and *Nv*Pbx (*Figure 8B,C*).

Together these results show that the NK-TALE and Hox-TALE interaction networks rely on different molecular properties, in particular with a role of Meis in promoting HX-dependent or HX-independent interaction modes, respectively (*Figure 8D*).

We next analysed the interaction properties between TALE proteins and the *Drosophila* Engrailed (En) protein, a member of the extended-Hox family, which was recently described to form cooperative DNA-binding complexes with Exd and Hth on physiological target sequences (*Fujioka et al., 2012*). We observed that En could form dimeric or trimeric complexes with Exd or Exd/Hth respectively, but in all cases these complexes were lost upon the mutation of the HX (*Figure 8—figure supplement 1*).

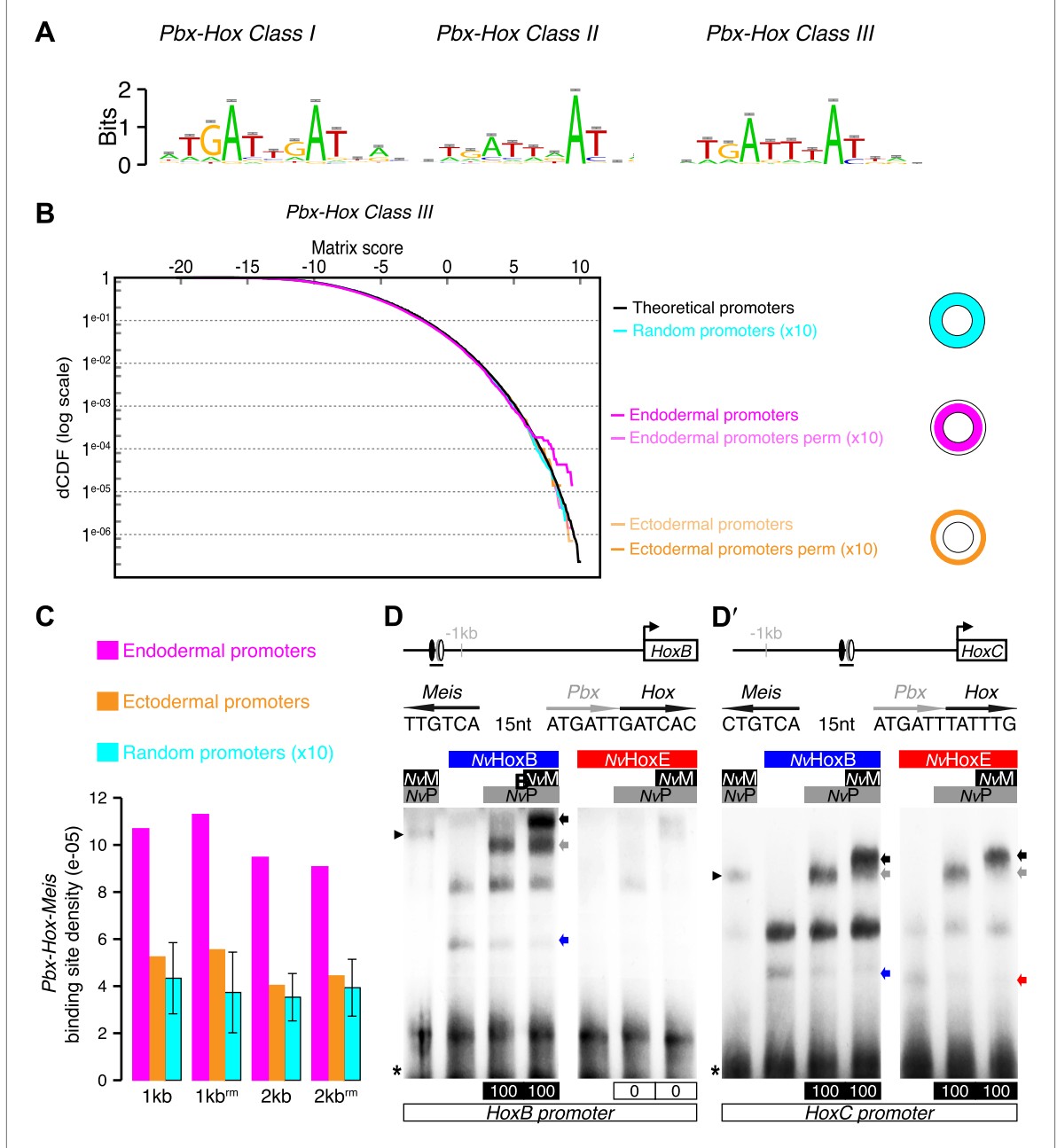

**Figure 5**. Genes expressed in the endoderm are enriched in Hox-TALE binding sites in their promoter region. (**A**) Hox/Pbx binding motifs represented as logos. The three motifs represent the binding specificity of the Hox/Pbx complex for sites of class I, II, or III. Matrix was determined by Selex with the Drosophila proteins (***Slattery et al., 2011***). (**B**) Score distributions of the Hox/Pbx Class III matrix. The Y-axis is shown in logarithmic scale to highlight the relevant range of p values (small values). The separation of the pink curve (endoderm) from the black one (theoretical distribution) indicates an enrichment of the Hox/Pbx putative binding sites in the promoter of genes expressed in the endoderm. On the contrary, there is no enrichment in the promoter of genes expressed in the ectoderm, as the orange curve follows the black one. All negative controls also show no enrichment: random sets of gene promoters (cyan), promoter regions randomized by matrix column permutations for the endoderm (light pink) and ectoderm (light orange). (**C**) In silico analysis of Hox/Pbx/Meis binding sites in the promoter region (1 kb or 2 kbs upstream of the transcription start site) of genes expressed in the endoderm (pink), ectoderm (orange), or randomly chosen (cyan). The graph illustrates the preferential enrichment of Hox/Pbx/Meis binding sites in the promoter region of endodermal genes. Rm: repeat masked. (**D–D′**) Band shift experiments between *Nv*Hox and *Nv*TALE proteins on binding sites found in the promoter region of *NvHoxB* and *NvHoxC* genes. Sequence and genomic position of each binding site are shown above the gel. Colour code and annotations are as in ***Figure 3***. Note the distinct DNA-binding preferences of *Nv*HoxB and *Nv*HoxE on these two different target sites. See also ***Figure 5—figure supplements 1 and 2***.

*Figure 5. Continued on next page*

*Figure 5. Continued*

The following figure supplements are available for figure 5:

**Figure supplement 1**. Genome-wide analysis cannot reveal significant enrichment of Hox/PBC/Meis binding sites.

**Figure supplement 2**. *Nv*Meis promotes HX-independent interaction modes on DNA-binding sites found in the promoter region of *NvHoxB* (A) and *NvHoxC* (B).

In conclusion, although *Nv*Msx and *Dm*En do not exhibit identical cooperative DNA-binding properties with PBC and Meis, they both require the HX to interact with the TALE partners (*Figure 8D*). Thus, the acquisition of the HX motif during evolution was likely a key molecular event for the emergence of ANTP–TALE interaction networks. Along the same line, we observed that the Trox2/Gsx protein from *Trichoplax adhaerens*, which does not contain any obvious HX-like sequence (*Schierwater and Kuhn, 1998*), is not able to form any dimeric or trimeric complex with PBC or PBC/Meis, respectively (*Figure 9A*).

To further confirm that ANTP-TALE networks are a metazoan innovation, we analysed the interaction properties of PBC and Meis proteins of *Acanthamoeba castellanii (Ac)*, a unicellular organism from the Amoebozoa group (*Figure 1*). Interestingly, *Ac*Meis possesses a MEIS-A domain and displays a high level of sequence similarity with mouse or fly Meis proteins in the HD (*Figure 2—figure supplement 2*). In contrast, *Ac*Pbx lacks any PBC-A domain and has a strongly divergent HD when compared to other Pbx proteins (*Figure 2—figure supplement 1*). As expected, we observed that *Ac*Pbx could neither bind on a consensus PBC-binding site, nor stimulates the binding of *Ac*Meis on the same probe, suggesting that both proteins could not interact in vitro (*Figure 9B*). This result was confirmed by using central or posterior Hox/PBC/Meis binding sites, on which no dimeric or trimeric complex with Hox proteins could be formed (*Figure 9C,D*). To assess whether absence of protein complexes could be explained by the strong sequence divergence of *Ac*Pbx, we repeated experiments with the mouse Pbx1 protein. Under these heterologous partnership conditions, we observed that the binding of *Ac*Meis could be strongly enhanced in the presence of the Pbx partner, suggesting that the two proteins could make interactions (*Figure 9B*). However, neither dimeric nor trimeric complexes could be formed on central and posterior Hox/PBC/Meis nucleotide probes, demonstrating that *Ac*Meis lacks protein feature(s) for making cooperative DNA-binding complexes with Hox and PBC proteins (*Figure 9C'–D'*). Not surprisingly, *Ac*Meis is also not able to rescue Hox/PBC complex formation upon the HX mutation (*Figure 9—figure supplement 1*).

Taken together, our results show that the use of other protein motifs than the HX for interaction with TALE partners is a peculiar property of Hox proteins among the ANTP class. They also emphasize that the PBC/Meis partnership likely evolved concomitantly with the apparition of the HX in the ANTP class. In this context, our work with TALE proteins of *Acanthamoeba* underlines that the evolution of the TALE partners enabled the interaction network with Hox proteins and hence new functions to emerge during eukaryote evolution.

## Discussion

### Molecular evolution of the HX motif

We have shown that identical molecular rules and conserved functions characterize Hox-TALE interaction networks in Cnidaria and Bilateria. The presence of the HX motif in several *Nv*Hox members and its requirement for generic Hox/TALE functions strongly suggest that this motif had a pivotal role for the evolution of an active Hox-TALE system in early metazoan lineages. In addition, the observation that a number of cnidarian Hox proteins do not have any HX (*Figure 10*, *Figure 10—figure supplement 1*) highlights that the ancestral molecular properties and hence functions of the Hox-TALE network could have considerably diverged among different cnidarian lineages.

To date, no HX motif can be found in any member of the ANTP class in Porifera, Ctenophora, or Placozoa group. This motif is present in different ANTP subclasses, including NK, Hox/ParaHox and extended-Hox, specifically in Bilateria and Cnidaria. Different scenarios can be proposed for explaining the evolutionary history of the HX among different ANTP members. These scenarios are hypothetical and diverge according to the putative evolutionary history of the ANTP class homeobox

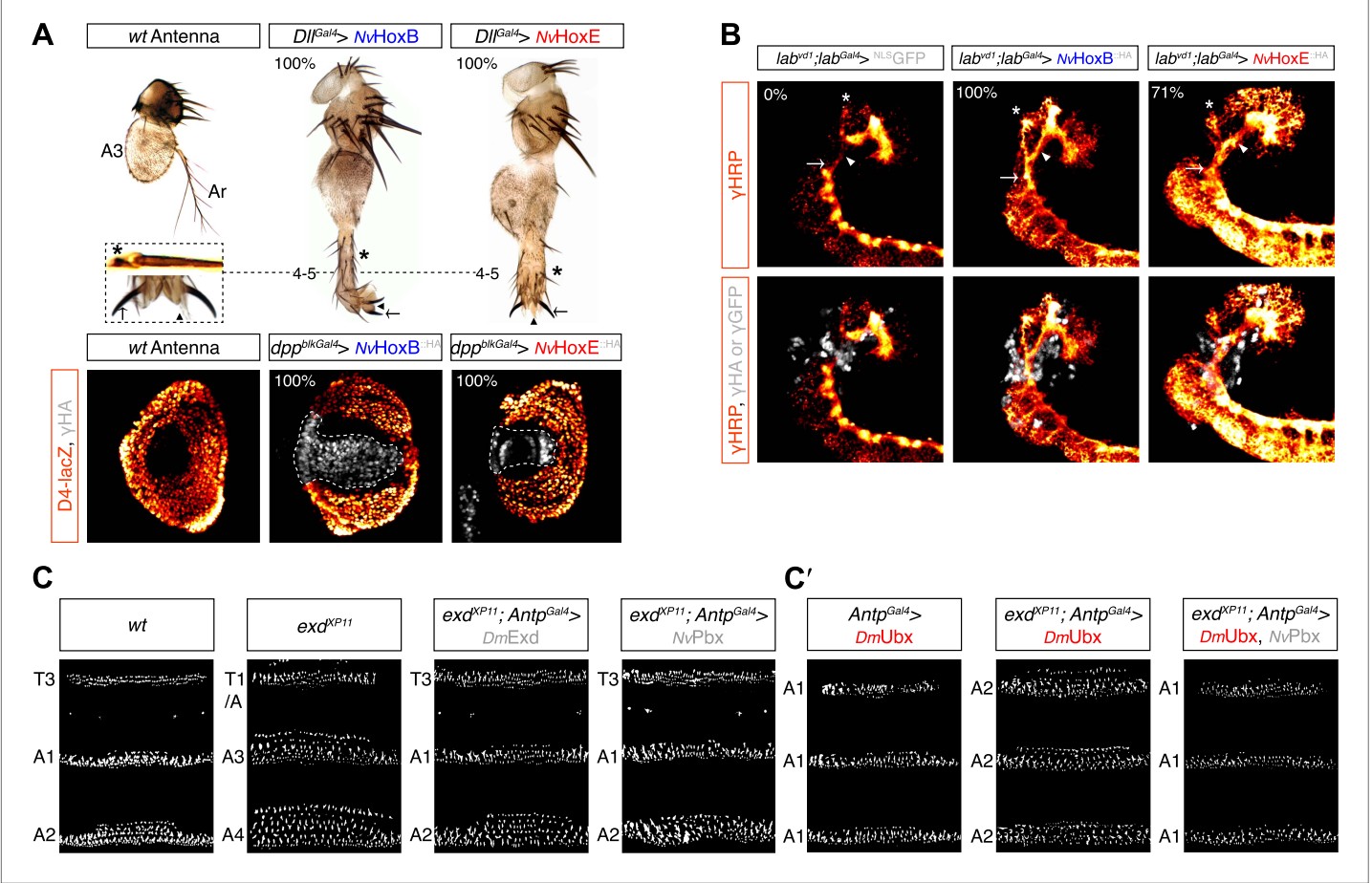

**Figure 6**. Functional analysis of *Nv*Hox and *Nv*Pbx proteins in Drosophila. (**A**) Antenna-to-leg transforming activities of *Nv*HoxB and *Nv*HoxE. *Nv*Hox proteins were expressed in the antenna with the *Distalless (Dll)-Gal4* driver. Asterisk depicts leg-specific bracted bristles. 4–5 shows the transformation of the arista in two tarsal segments. Arrow and arrowhead in the enlargement indicate the formation of the leg-specific terminal claw and its associated sensory pad respectively. The antenna-to-leg transformation by *Nv*Hox proteins (grey) is achieved through the repression of the *spineless* (*ss*) target gene, as observed by the repression of the *ss* enhancer *D4* activity on *lacZ* reporter gene expression (orange). See also **Figure 6—figure supplements 1 and 2**. (**B**) Rescue of the *labial* (*lab*) mutant phenotype in the tritocerebrum by *Nv*Hox proteins. The central nervous system is stained with an anti-HRP (orange). Hox or GFP (as a control) proteins (grey) are expressed in the tritocerebrum with a *lab-Gal4* driver. Frontal connectives (asterisk), longitudinal connectives (arrowhead) and tritocerebral commissure (arrow) are indicated. In *lab* mutant background, longitudinal connectives are reduced, frontal connectives project ectopically and the tritocerebral commissure is missing (**Hirth et al., 1998**). Expression of *Nv*HoxB or *Nv*HoxE in this mutant context leads to a complete or strong rescue of this phenotype, respectively. See also **Figure 6—figure supplement 2**. (**C–C′**) *Nv*Pbx can rescue zygotic *exd* mutant phenotypes in the Drosophila larva cuticle. (**C**) Larvae homozygous for the zygotic *exd^XP11* mutation have T3 and A1 segments that resemble to a T1/abdominal or A3 segment, respectively. Thoracic expression of either *Dm*Exd or *Nv*Pbx in this mutant background (through the UAS/Gal4 system, with the *Antennapedia (Antp)-Gal4* driver) is sufficient to restore the correct specification of T3 and A1, as assessed by the shape and arrangement of denticle belts. (**C′**) Ubx normally specifies the A1 segment. Ectopic expression of Ubx with *Antp-Gal4* induces A1-like segments in the thorax. In absence of Exd, Ubx produces A2-like segments. Providing back *Nv*Pbx in this genetic background is sufficient to restore the normal A1-inducing activity of Ubx. See also **Figure 6—figure supplements 3 and 4**.

The following figure supplements are available for figure 6:

**Figure supplement 1**. *Nv*HoxB and *Nv*HoxE interact with the Drosophila TALE cofactors Extradenticle (Exd) and Homothorax (Hth) in vitro and in vivo.

**Figure supplement 2**. Role of the HX of *Nv*Hox proteins in generic Drosophila Hox assays.

**Figure supplement 3**. *Nv*Pbx interacts with the Drosophila Ultrabithorax (Ubx) and Homothorax (Hth) proteins in vitro and in vivo.

**Figure supplement 4**. *Nv*Meis reproduces generic bilaterian Meis activities in the Xenopus embryo.

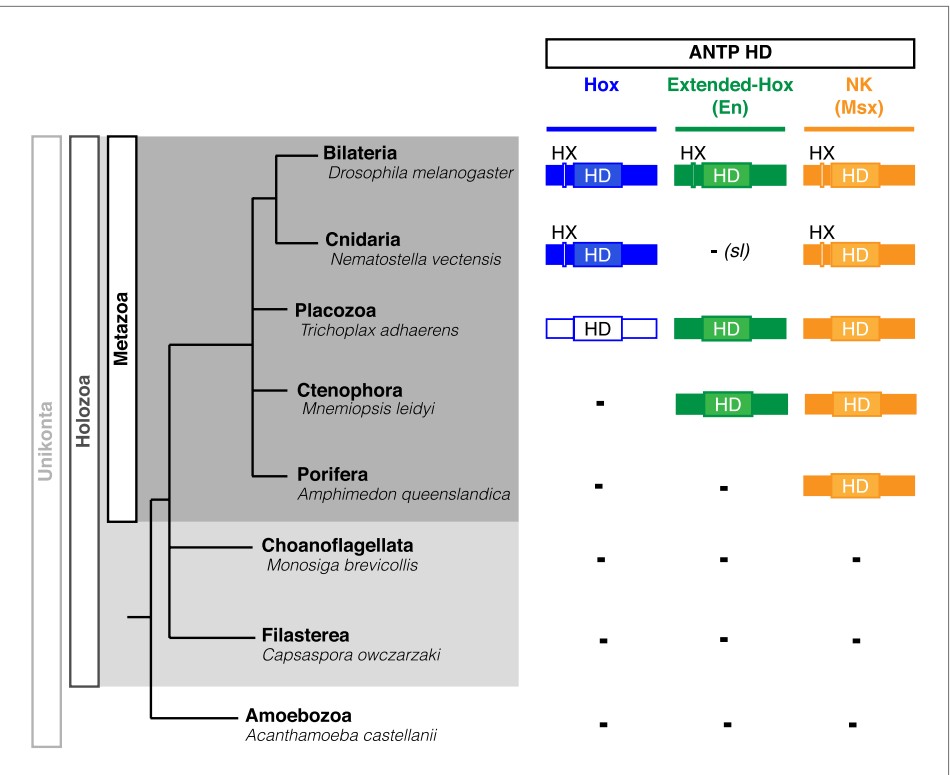

**Figure 7**. Phylogeny of Hox, Extended-Hox (Engrailed, En) and NK (Msx) proteins during eukaryote evolution. Nomenclature is as in *Figure 1*. The conserved features of any HX motif correspond to a sequence containing an invariant tryptophan residue in a hydrophobic context, with a lysine or arginine residue at position +2 to +5, as previously defined (*In der Rieden et al., 2004*).

genes. In one scenario, HX-containing NK, Hox and extended-Hox members could have emerged from a common HX-containing NK protein, which itself appeared from duplications of an ancestral HX-deficient NK cluster, at the basis of Eumetazoa (Bilateria+Cnidaria, *Figure 11A*). Alternatively, it was recently proposed that Hox/ParaHox, NK and extended-Hox evolved before the origin of poriferans (*Mendivil Ramos et al., 2012*), being already present in the last common ancestor of animals (Urmetazoa). In this scenario, we do not favour the hypothesis of independent acquisitions of the HX in the three homeobox gene families. Indeed, although the HX is a short motif, it is always located upstream and at a reasonable distance of the HD. This invariant position is probably critical to ensure interactions with PBC proteins. Moreover, the distance between the HX and the HD could have played important roles for the acquisition of new functions, as noticed in certain bilaterian Hox proteins (*Prince et al., 2008*; *Saadaoui et al., 2011*). Thus, the HX could already have been present in the ProtoANTP protein of the Urmetazoa ancestor, being secondarily lost in Porifera, Ctenophora and Placozoa during evolution (*Figure 11B*).

## Hox/TALE functions in the sea anemone *Nematostella vectensis*

Our results showed that *Nematostella* Hox and TALE proteins shared conserved functions with their bilaterian counterparts. Interestingly, *Nv*HoxB and *Nv*HoxE display anterior-like or central-like properties, respectively. These preferential activities were observed in vitro, at the level of DNA-binding site recognition, but also in the generic rescue assay of the *Drosophila Hox*-mutant tritocerebrum structure. Thus, although *Nv*HoxB and *Nv*HoxE are not organized in a genomic cluster, they display differential expression profiles and activities, suggesting that ancestral colinearity rules are at least kept for two asymmetrically expressed Hox genes in *Nematostella*.

Considering our in silico data, and given the expression profile of Hox and TALE members in the *Nematostella* embryo, we propose that Hox–TALE interaction networks could be used for regulating gene expression in the endoderm, likely for positioning and specifying the formation of the different

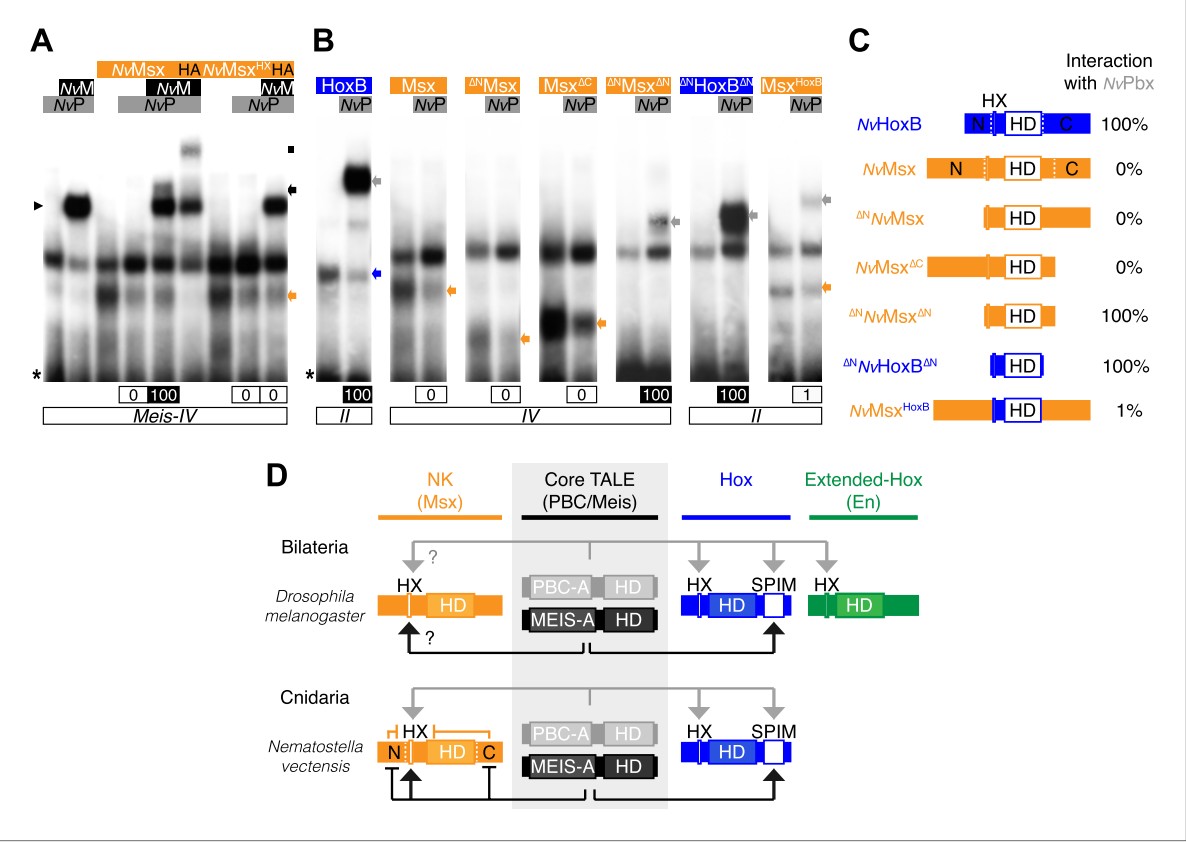

**Figure 8**. Genesis of Hox–TALE interaction networks in Metazoa. (**A**) Band shift experiments between *Nv*Msx and *Nv*TALE cofactors, as indicated. Colour code and annotations are as in previous figures. Note that no dimeric *Nv*Msx/*Nv*Pbx complex is formed. Binding reactions with *Nv*Msx proteins were performed on a consensus *Msx/Pbx* binding site derived from vertebrates and containing an additional *Meis* binding site in a topology similar to the Hox probe ('Materials and methods'). (**B**) Band shift experiments between wild-type, truncated or chimeric *Nv*Msx and/or *Nv*HoxB proteins, and *Nv*Pbx, as indicated. (**C**) Scheme of the diverse protein constructs and their corresponding interaction affinity level with *Nv*Pbx, as assessed from quantification of each band shift. Quantifications with truncated *Nv*Msx proteins were deduced by comparison with the trimeric *Nv*Msx/*Nv*Pbx/*Nv*Meis in (**A**). (**D**) Molecular rules underlying interaction properties between NK, Hox, or extended-Hox members and TALE cofactors in Eumetazoans. In this model, Meis is able to promote HX-dependent interactions between *Nv*Msx and *Nv*Pbx by masking inhibitory interaction domains in *Nv*Msx. Whether a similar role could exist in Bilateria remains to be determined. We noticed that the Drosophila Msx protein contains an HX and forms trimeric but not dimeric complexes with the Drosophila TALE partners (not shown). HX-dependency in those interactions remains to be determined (question marks). Interaction between *Dm*En and PBC is also HX-dependent but does not require the presence of Meis to occur. In contrast to the NK or extended-Hox families, most members of the Hox family have retained a HX motif. This motif is required for generic Hox/Pbx functions. The additional presence of Meis allows revealing specific Pbx interaction motifs (SPIMs), which could be important for distinguishing and/or diversifying the embryonic activities of each Hox paralog group member. See also *Figure 8—figure supplement 1*.

The following figure supplements are available for figure 8:

**Figure supplement 1**. The Drosophila Engrailed (*Dm*En) protein forms HX-dependent DNA-binding complexes with Exd (*Dm*E) and Hth (H) on physiological target sites.

mesenteries along the directive axis. Homology between cnidarian and bilaterian axes is a long-standing and still controversial question. It was first proposed that the oral–aboral axis of *Nematostella* could be orthologous to the bilaterian AP axis, with an anterior-like (*NvHoxA*/*anthox6*) and central/posterior (*NvHoxF*/*Anthox1*) Hox gene being expressed at the oral or aboral tip respectively (**Finnerty et al., 2004**). This expression profile is however not conserved in other cnidarian species (**Kamm et al., 2006**) and there is compelling evidence that the aboral pole could rather correspond to the anterior end of bilaterians (**Sinigaglia et al., 2013**). Therefore, expression of *Nv*Hox genes along the primary body axis could correspond to individual morphogenetic and not positional patterning functions, as recently shown for *NvHoxF* (**Sinigaglia et al., 2013**). In this context, other HD-containing determinants could

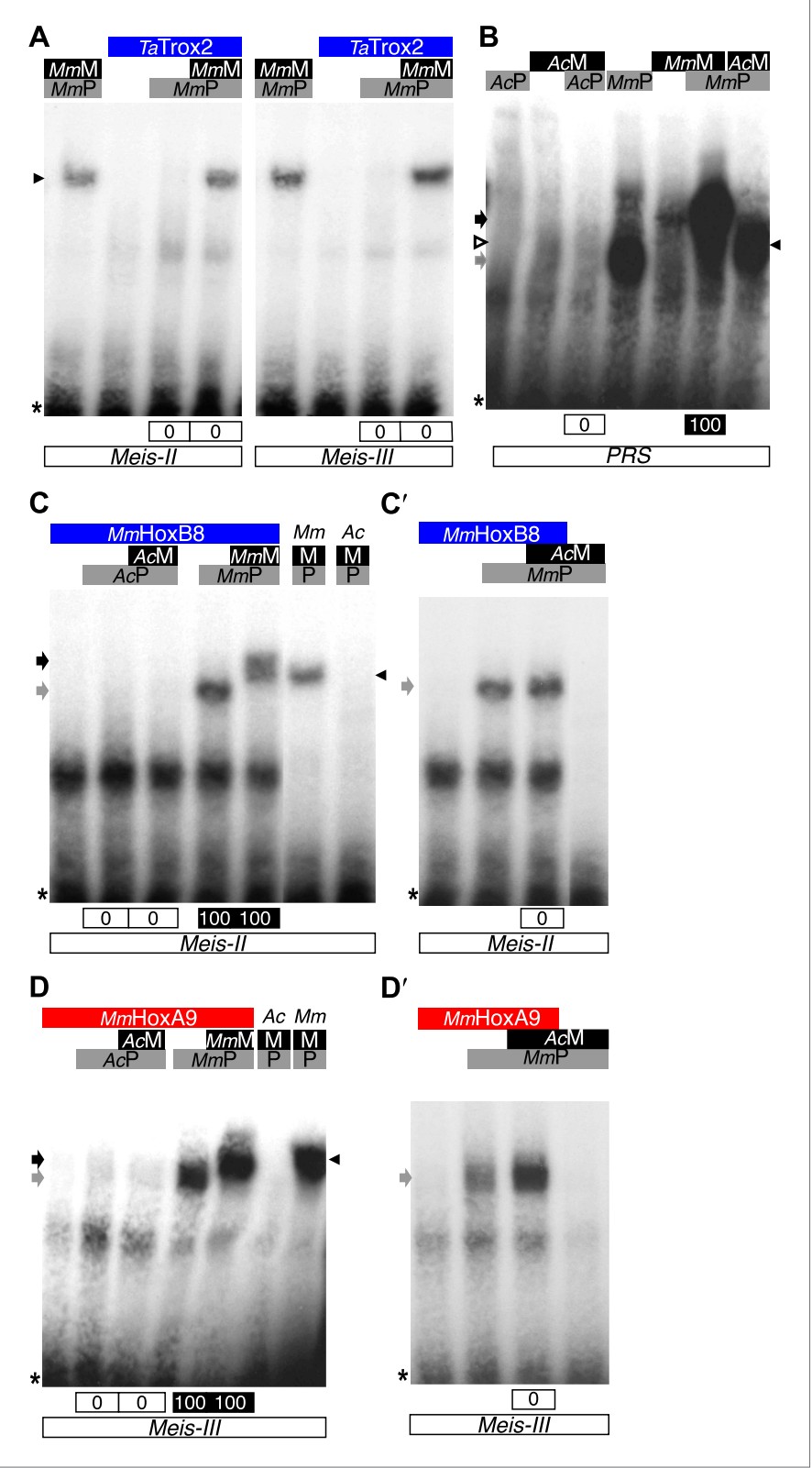

**Figure 9**. Interaction properties of Hox and TALE proteins from *Trichoplax adhaerens* and *Acanthamoeba castellanii*. (**A**) The ProtoHox/ParaHox Trox2 protein from *Trichoplax adhaerens* does not form DNA-binding complexes with PBC or PBC/Meis in vitro. Band shift experiments are performed with mouse Pbx (*Mm*P) and Meis (*Mm*M) proteins on central (*Meis-II*) and posterior (*Meis-III*) Hox/PBC/Meis binding sites as indicated. Black

*Figure 9. Continued on next page*

*Figure 9. Continued*

arrowhead shows dimeric Pbx/Meis complexes. (**B**–**D'**) PBC (*Ac*P) and Meis (*Ac*M) proteins from the unicellular *Acanthamoeba castellanii* organism cannot form protein complexes between each other or with Hox proteins. (**B**) Band shift experiment on a consensus PBC binding site (*PRS*, ***Chang et al., 1995***). *Ac*P does not bind DNA, neither as a monomer nor with *Ac*M. A weak monomer DNA-binding of *Ac*M is observed (white arrowhead). This monomer binding is strongly enhanced in the presence of mouse Pbx1 (*Mm*P, black arrowhead). In comparison, *Mm*P binds strongly (grey arrow), and the monomer binding of Meis1 (*Mm*M) is also strongly enhanced in the presence of Pbx1 (black arrow). (**C**–**C'**) Band shift experiments with mouse HoxB8 (*Mm*HoxB8) and mouse or *Acanthamoeba* TALE cofactors on the central (*Meis-II*) Hox consensus binding probe as indicated. (**D**–**D'**) Band shift experiments with mouse HoxA9 (*Mm*HoxA9) and mouse or *Acanthamoeba* TALE cofactors on the posterior (*Meis-III*) Hox consensus binding probe as indicated. Complexes with Hox proteins are observed only with mouse Pbx (grey arrows) and Pbx/Meis partners (black arrows) on both probes. *Ac*Pbx and *Ac*Meis proteins are not able to form dimeric complexes on these probes, unlike mouse TALE proteins (black arrowheads). See also ***Figure 9—figure supplement 1***.

The following figure supplements are available for figure 9:

**Figure supplement 1**. PBC and Meis proteins from the unicellular *Acanthamoeba castellanii* organism do not form protein complexes with mouse HX-mutated Hox proteins.

play important patterning roles (***de Jong et al., 2006***), as demonstrated for Six transcription factors in the early specification of the aboral pole (***Sinigaglia et al., 2013***).

Along the same line, the directive axis of *Nematostella* was proposed to be homologous to the bilaterian DV axis (***Matus et al., 2006***), but functional analyses revealed that DV patterning genes are also required in the endoderm and ectoderm along the primary axis (***Matus et al., 2006***; ***Saina et al., 2009***).

Together, these observations highlight that patterning molecules can be used along different longitudinal axes during animal evolution, which renders difficult the comparison between Bilateria and Cnidaria. Here, we propose that the ancestral molecular cues underlying the Hox patterning system along the cnidarian directive axis could have been recruited for AP patterning in Bilateria.

## TALE proteins and the evolution of interaction networks in eukaryotes

Non-TALE and TALE representatives of the HD superfamily were probably already present in first eucaryotes (***Derelle et al., 2007***; ***Larroux et al., 2008***; ***Ryan et al., 2010***), and it was proposed that interactions between these two classes of TFs could have existed in the common ancestor of plants, fungi, and metazoans (***Burglin, 1998***). TALE proteins originate from a putative ancestor that contained a MEINOX domain and gave rise to conserved N-terminal interaction domains in different TALE members (***Burglin, 1997***). Interestingly, although PBC and Meis are not present outside Unikonta, other TALE members are known to interact with each other or with other protein families in plants (***Bellaoui et al., 2001***; ***Hackbusch et al., 2005***; ***Kanrar et al., 2006***; ***Hay and Tsiantis, 2010***) and fungi (***Keleher et al., 1989***; ***Stark and Johnson, 1994***; ***Li et al., 1998***; ***Carr et al., 2004***), suggesting that partnership with TALE proteins is a common and ancient feature in eukaryotes (***Figure 11C***; ***Burglin, 1998***).

Here, we revealed the existence of interactions between typical HD (ANTP) and TALE (PBC/Meis) members in Cnidaria, suggesting that this network, which is also present in Bilateria, was already at work in the Eumetazoa ancestor, before the Cnidaria/Bilateria split. We showed that more ancient TALE proteins, like those from the unicellular *Acanthamoeba* organism could neither interact between each other nor form complexes with Hox proteins. Heterologous interaction assays between *Ac*Meis and mouse Hox and PBC proteins further exemplified that the PBC/Meis partnership is critical for the formation of Hox–TALE networks. This partnership probably appeared with the PBC-A domain in PBC, and concomitantly with the HX motif during eukaryote evolution. Although the MEIS-A domain was more ancient, the Meis partner also clearly acquired additional protein features, allowing the formation and therefore diversifying the activity of Hox/PBC/Meis networks in Metazoa.

Interestingly, the HX is not only conserved in the Hox family but also in different NK and extended-Hox members. Still, the role and the importance of TALE partners in these additional networks might not be equivalent. We propose that apparition of Hox, NK and extended-Hox members was accompanied by functional sub-specialisations that could in part result from divergent molecular interaction properties with TALE cofactors. For example, only two NK sub-family members (Msx and Tlx) have

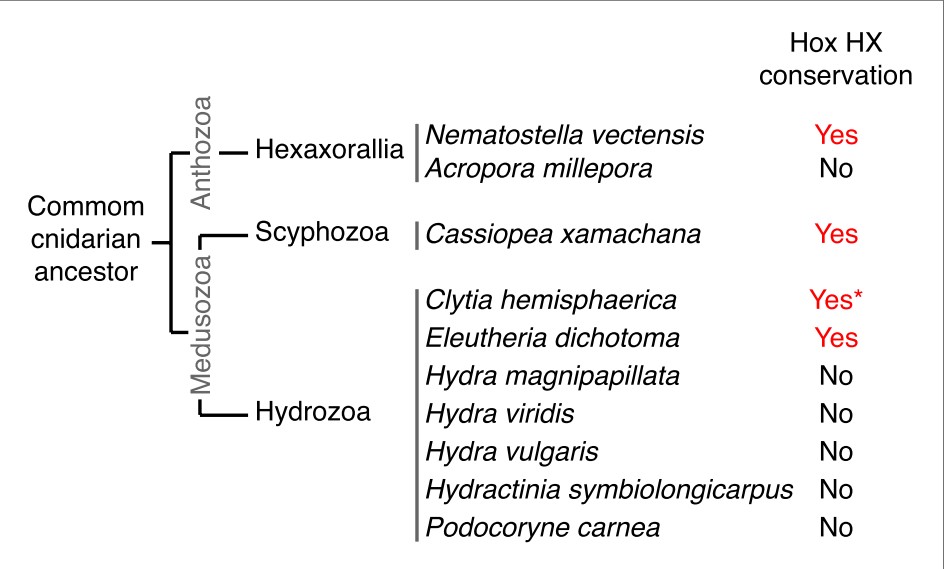

**Figure 10**. Phylogeny of the HX in Hox proteins of main cnidarian lineages.
The following figure supplements are available for figure 10:

**Figure supplement 1**. Protein sequence alignment of the region encompassing the HX (highlighted in red) and HD (highlighted in yellow) of cnidarian Hox members used in *Figure 10—figure supplement 1*.

retained a HX motif, suggesting that interaction between NK and TALE proteins is not a general rule. The same rationale applies to extended-Hox members. On the contrary, the HX is present in almost all Hox paralog groups, which coincides with a general requirement of TALE cofactors in Hox functions. In this context, the role of Meis for revealing additional and more specific Pbx interaction motifs (SPIMs) in Hox proteins (*Hudry et al., 2012*) was also likely an ancestral feature of the Hox/TALE system for distinguishing functions between different Hox paralog members.

In conclusion, we propose that Hox-TALE networks constituted an ancestral regulatory module that was later on exploited for patterning functions in Bilateria. This network was effective as soon as different interaction modes could exist with duplicated Hox family members, allowing diversifying patterning functions along the body axis. This original molecular system was subsequently co-opted by the various contexts of embryogenesis in different eumetazoan phyla, for axis or tissue (see e.g., *Di-Poi et al., 2007*) patterning, illustrating its remarkable adaptability throughout animal evolution.

## Materials and methods

### Cloning
Clones were generated by PCR from full-length complementary DNAs and restriction-cloned in the appropriate vector (see also *Supplementary file 1* for a complete list of all the constructs). Primers used are listed in *Supplementary file 1*. All constructs were sequence-verified before using.

### Fly stocks and transgenic lines
Transgenic lines were established either by the PhiC-31 integrase system (with the pUASTattB vector [*Venken et al., 2006*; *Bischof et al., 2007*]) or by classical P-element (with PUAST vector) mediated germ line transformation. Unless otherwise indicated, fly stocks were obtained from the Bloomington *Drosophila* Stock Center. Gal4 drivers used are: *en*-Gal4, *Dll*-Gal4, *Ubx*-Gal4$^{M1}$ (kindly provided by Ernesto Sanchez-Herrero), *Antp*-Gal4 (Michel Crozatier), and *dpp*$^{blink}$-Gal4. *lab*$^{VD1}$; *lab*-Gal4 line was provided by Frank Hirth, and *D4*-LacZ line by Ian Duncan.

### Immunostainings, cuticle preparation and in situ hybridization
Immunodetections in *Drosophila* embryos, imaginal discs, and cuticle preparations were performed according to standard procedures. *Nematostella* in situ hybridizations were performed as described

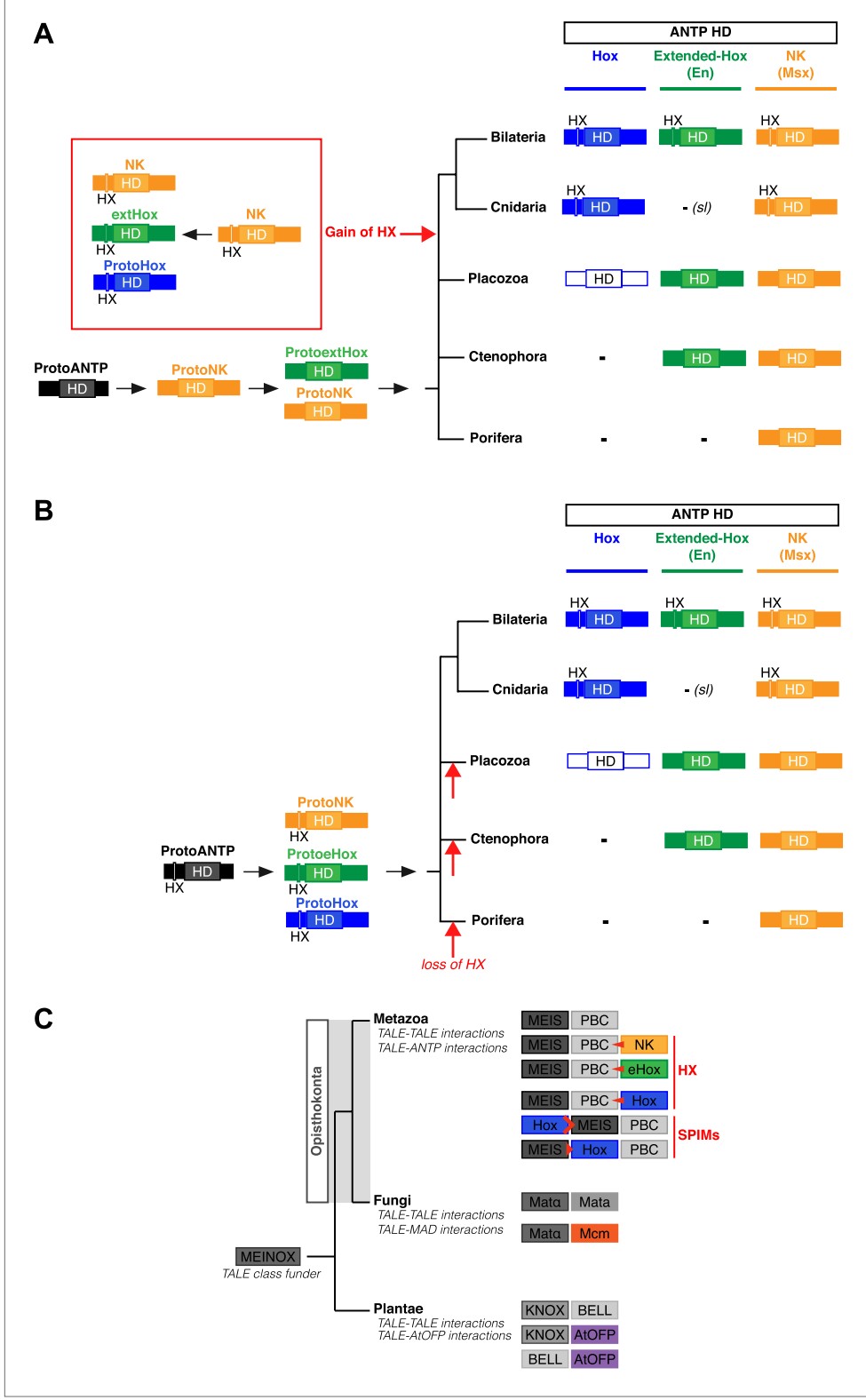

**Figure 11**. Evolution of molecular signatures underlying interaction networks with TALE proteins. (**A**) Model for the apparition of the HX in ANTP members of Cnidaria and Bilateria. In this scenario, the HX appeared in a duplicated NK protein that gave rise to other HX-containing ANTP members in the common ancestor of Bilateria and Cnidaria. Duplications in other lineages occurred without the apparition of the HX. (**B**) Second model for the apparition of
*Figure 11. Continued on next page*

*Figure 11. Continued*

the HX in ANTP members of Cnidaria and Bilateria. This model is based on a recent work postulating the existence of Hox/ParaHox, NK, and extended-Hox clusters in the common ancestor of animals (*Mendivil Ramos et al., 2012*). In this scenario, it is unlikely that the HX appeared after duplications of the ProtoANTP ancestor independently and at the same place in the ProtoHox, ProtoNK and ProtoExtended(e)Hox families. Thus, the HX motif was probably already present in the ProtoANTP ancestor. This motif was secondarily lost in ANTP members of Porifera, Placozoa and Ctenophora during evolution. See also 'Discussion'. (**C**) TALE proteins and the generation of interaction networks across major multicellular branches of the eukaryote evolutionary tree. TALE proteins are indicated in grey-filled boxes, with different grey tones corresponding to different TALE members. Other colours depict non-TALE members. Interactions can occur between different TALE members, or between TALE and non-TALE members. These interactions involve different proteins in each major multicellular branch, as indicated. Red signs in Metazoa symbolise interaction modes involving the HX or specific PBC interaction motifs (SPIMs) between TALE (PBC/Meis) and ANTP (NK, extended(e)Hox, Hox) members.

(*Genikhovich and Technau, 2009*). The antibodies used were: rat anti-HA (1/500; Molecular probe, Invitrogen, CA, USA), mouse anti-β-galactosidase (1/500; Molecular Probe), chicken anti-GFP (1/500; Promega, WI, USA), guinea pig anti-Homothorax (Natalia Azpiazu) and rabbit anti-HRP (1/100; FITC-conjugated, Jackson Immunoresearch, PA, US).

## BiFC analysis in *Nematostella* and *Drosophila* embryos

BiFC analysis in *Drosophila* embryos was performed as previously described (*Hudry et al., 2011*). BiFC in *Nematostella* was achieved by injecting in vitro synthesized mRNAs. mRNAs synthesis was performed on a template produced by PCR using the mMessage mMachine T7 Kit (Ambion, Invitrogen, CA, USA). Embryos were co-injected with two BiFC vectors (50 ng/µl each) and fluorophore coupled-dextran. Embryos were allowed to develop for 24 hr and were processed for visualisation.

## Electro-mobility shift assays

Constructs for EMSAs were cloned in the pCDNA3 vector and sequence-verified before using. Proteins were produced with the TNT-T7-coupled in vitro transcription/translation system (Promega, WI, USA). Production yields of wild-type and mutated counterpart proteins were estimated by $^{35}$S-methionine labelling. EMSAs were performed as described previously. We used the following double strands radiolabelled probes: *Class I* 5'-ACGCGGGAATGATTGATGGCCCAAATA, *Meis-II* 5'-ATGACAGCTCGGAATGATTAATGGCCCAAATA, *Meis –IV* 5'- ATGACAGCTCGGAATGATTAATTACCCAAATA *a1a promoter* 5'-TAATATTGTCAGTCAGATTGCAAATGATGATTGATCACTATAG, *a7 promoter* 5'-TAGGTCTGTCAGGCTGCTCTTTCACGATGATTTATTGCCTCAC, box2' from the tsh epidermal enhancer *tsh* 5'-TCATGGACTGAAAACCATAAATTTGATAATTGACTTTCCAC (*McCormick et al., 1995*), *DllR* 5'-TATTTGGGAAATTAAATCATTCCCGCGGACAGTT (*Gebelein et al., 2002*), *D4$^{1–62}$* 5'-AGTTTACCATTAAATTCCCATTTAGGCTGTCAATCATTTGCGCTAATTTTTCTTGGCGGCTT (*Duncan et al., 2010*), *class IV* 5'- ATGACAGCTCGGAATGATTAATTACCCAAATA, *lab$^{48/95}$* 5'-AAATTGATGGATTGCCCGGCGCCGACTGTCACCG (*Ryoo et al., 1999*), and *modC site I* 5'-CCTCGTCCCACAGCTATAATGATTAATGAACGCGCCGCC (*Joshi et al., 2010*). The sequence of all other probes (*probes class II and III*) is indicated in the corresponding figures. 1 mm of rat anti-HA (1/50; Molecular probe, Invitrogen, CA, USA) was used for the 'super-shift' experiments. Quantifications of shifted bands was performed using the Analyze>Gels function of the ImageJ software.

## Bioinformatic analysis

By screening the literature two sets of 38 genes exclusively expressed in the endoderm or the ectoderm of *Nematostella* embryo could be identified. The density of the Pbx-Hox and Meis motifs in the *cis*-regulatory regions of these 72 genes was determined, using a previously established matrix (*Slattery et al., 2011*). For more details, see the following website: http://www.bigre.ulb.ac.be/Users/morgane/bruno/result.html.

## *Xenopus* assays

One-cell stage *Xenopus laevis* embryos were microinjected with RNA encoding *Nematostella vectensis* Meis (NvMeis, 0.8 ng) protein. Animal Cap (AC) explants were removed from control and

injected embryos at blastula stage and cultured to neurula stage 16. Total RNA was isolated from five control embryos (CE) and eighteen ACs from the control or injected groups. Semi-quantitative (sq) RT-PCR analysis was performed to posterior neural markers as described (*Elkouby et al., 2010*). EF1alpha controls for RNA levels in each sample. RT-PCR was performed on total RNA from embryos.

## Acknowledgements

We warmly thank Peter Holland, Michael Manuel, Michalis Averof, Vincent Laudet, and Abderrahman Khila for critical comments and Johanne Burden for proofreading the manuscript. We thank Jun Aruga, Ian Duncan, Natalia Azpiazu, Bruno Bello and the Bloomington stock centre for reagents and fly lines. We thank Yacine Graba for allowing us performing part of this work in his laboratory. Research in the author's laboratory is supported by the CNRS, Ecole Normale Supérieure (ENS) of Lyon, EMBO short-term fellow (for B Hudry) and grants from ARC (Association de Recherche contre le Cancer) and FRM (Fondation pour la Recherche Médicale).

## Additional information

### Funding

| Funder | Author |
|--------|--------|
| Association pour la Recherche contre le Cancer | Bruno Hudry, Morgane Thomas-Chollier, Yael Volovik, Marilyne Duffraisse, Amélie Dard, Dale Frank, Ulrich Technau, Samir Merabet |
| Fondation pour la Recheche Medicale | Bruno Hudry, Morgane Thomas-Chollier, Yael Volovik, Marilyne Duffraisse, Amélie Dard, Dale Frank, Ulrich Technau, Samir Merabet |
| EMBO short term fellow | Bruno Hudry |

The funders had no role in study design, data collection and interpretation, or the decision to submit the work for publication.

### Author contributions

BH, SM, Conception and design, Acquisition of data, Analysis and interpretation of data, Drafting or revising the article; MT-C, Acquisition of data, Analysis and interpretation of data, Drafting or revising the article; YV, MD, AD, DF, Acquisition of data, Analysis and interpretation of data; UT, Analysis and interpretation of data, Drafting or revising the article

## Additional files

### Supplementary files

• Supplementary file 1. Constructs generated for BiFC, functional and shift assays.

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
