## [Decision Letter]

Thank you for sending your work entitled “Molecular insights into the origin of the Hox-TALE patterning system” for consideration at *eLife*. Your article has been favorably evaluated by a Senior editor and 3 reviewers, one of whom is a member of our Board of Reviewing Editors, and one of whom, David Ferrier, has agreed to reveal his identity.

The Reviewing editor and the other reviewers discussed their comments before we reached this decision, and the Reviewing editor has assembled the following comments to help you prepare a revised submission. 

The comments below are a recombined text from the major comments provided by the three reviewers. All three support publication, but ask for some additional discussion and clarifications. This requires only modifications in the text, but no additional experiments. 

The authors provide compelling functional evidence for a conservation of the Hox-TALE interaction network between Cnidaria and Bilateria. This is based on in vitro and vivo studies, including rescue constructs across taxa. They provide also negative evidence for some unclear homologs from lower taxa. The function of the Hox genes in cnidarians is a particularly contentious issue within the field and this manuscript represents an important new perspective on this debate, now supported with abundant new functional data. The detailed functional experiments performed here, involving two different *Nematostella* Hox genes clearly illustrate that this cnidarian has Hox (and Pbx and Meis) genes that function like the Hox genes of bilaterians, regardless of the difficulties of comparing the embryology and axial development of cnidarians relative to bilaterians. The variety in the modes of interaction between the Meis, Pbx and Hox proteins when the presence of a hexapeptide is varied is also of widespread interest, not just for the evolutionary implications highlighted in this manuscript, but also for the basic mechanistic implications for these transcription factors. 

In conclusion, this paper provides some surprising results as to the functional conservation of Hox/PBC/Meis interactions. It tries to argue that the emergence of the HX motif in the superfamily has allowed a greater role for these proteins but it is difficult to see how the HX motif, as small as it is, could have evolved so many times. Yet, this represents a large amount of molecular and in vivo work that leads to the model that, by acquiring the ability to interact, the three protein classes have allowed the specification of positional information in animals. 

There are, however, a few points that need additional discussion or a better placement in the relevant context: 

1) The first difficulty relates to the orthology of Hox genes, or even their real nomenclature, which is difficult to assess in the absence of clustering and extensive sequence landmarks. However, the authors manage to define them as 'anterior' or more posterior. 

2) The authors use BiFC extensively for defining protein–protein interactions, although it is not clear how specific these interactions might be. 

3) The in silico search for mesodermal binding sites for Hox/Pbx (and Meis) is too biased and not terribly useful to show functional specificity. This only allows the definition of potential binding sites but the same data could have been obtained by using bona fide DNA sites. It is not clear what the authors are trying to say with these sites. 

4) The more novel part of the paper concerns the extended Hox protein family, including NK proteins that also share an HX motif in higher animals. It is difficult to interpret the lack of direct interaction between Msx and PBC, likely due to the absence of an HX motif. However, the authors argue that the HX motif has appeared late in the superfamily of Hox proteins, but this is difficult to reconcile with the evolution of these families since this would imply that the HX domain has evolved independently in multiple proteins, after the duplication of the precursor Hox. 

5) It seems that a bit more speculation would be warranted to place the work in context. For example, they do not touch the question of axis very much. In Figure 1 they represent the aboral–oral axis of Cnidaria analogous to the anterior-posterior in bilateria. In the same figure they show evidence for asymmetry of Hox gene expression, along the directive axis, as it was also found by others. This would of course be in line with the enterocoel hypothesis and their short remark in the text would also support this. Hence, I would encourage them to add a paragraph on discussing axis homologies, although I do not expect that they can arrive at a final conclusion on this yet. Still, their work evidently contributes to this important question. 

6) For the revisions, authors should revisit the question of which figures should be in the main text and which provide only supplementary information.

---

## [Author Response]

*1) The first difficulty relates to the orthology of Hox genes, or even their real nomenclature, which is difficult to assess in the absence of clustering and extensive sequence landmarks. However, the authors manage to define them as 'anterior' or more posterior*.

We agree there is a real difficulty in defining the orthology of cnidarian Hox genes. What is undisputed is the presence of anterior Hox genes. The others (in our terminology *NvHoxE/F*) are more difficult to place, as discussed in the first part of the result section. Another difficulty is the terminology of cnidarian Hox genes, which changes depending on the species and/or the authors. To be less confusing for non-cnidarian specialists, we now started the result section by providing a precise definition of the terminology and the possible orthology of *Nematostella* Hox genes. Figure 1 was modified accordingly and now depicts the genomic organisation of *Nematostella* Hox genes with their corresponding groups of orthology. We also provided protein sequence alignments between *Nematostella* and *Drosophila* Hox proteins to better highlight the degree of conservation at the level of the HX- and HD-encompassing regions (Figure 2—figure supplement 3 and Figure 2—figure supplement 4).

*2) The authors use BiFC extensively for defining protein–protein interactions, although it is not clear how specific these interactions might be*.

Specificity of BiFC is a critical point, especially when considering that the complementary fragments of the fluorescent protein have a certain affinity, leading to a stabilization of the interactions between two candidate fusion proteins. Therefore it is important to have negative controls in hands, to confirm that BiFC signals are relevant of real protein–protein interactions in vivo. We already described all suitable negative controls in *Drosophila*, mammalian cells and chick embryos (Hudry et al., BMC Biology 2011; Hudry et al., PLOS Biology 2012). These negative controls were repeated in our experiments with *Nematostella* fusion proteins. In particular, we showed that each individual sub-fragment of the Venus protein could not interact with the complementary fragment fused to *Nv*Pbx or *Nv*Meis (Figure 3). This is now more clearly stated in the Result section: “No BiFC was obtained between a fusion construct and the complementary isolated VC or VN fragment, highlighting that the interaction between *Nv*Pbx and *Nv*Meis fusion proteins was not artificially induced by the inherent affinity of the VN and VC fragments (Figure 3)”. Along the same line we provided evidence that BiFC cannot occur in conditions where Hox-PBC interactions are abolished (with the DNA-binding deficient form of *Nv*Pbx: Figure 3).

*3) The in silico search for mesodermal binding sites for Hox/Pbx (and Meis) is too biased and not terribly useful to show functional specificity. This only allows the definition of potential binding sites but the same data could have been obtained by using bona fide DNA sites. It is not clear what the authors are trying to say with these sites*.

We have now described more precisely the logic behind the search for *Nematostella* Hox/TALE binding sites in the genome: “Our band shift assays were performed on consensus binding sites previously defined with bilaterian Hox and TALE proteins. To know whether such sites could be used in the context of *Nematostella* development, we searched for their presence in the *Nematostella* genome.”

*4) The more novel part of the paper concerns the extended Hox protein family, including NK proteins that also share an HX motif in higher animals. It is difficult to interpret the lack of direct interaction between Msx and PBC, likely due to the absence of an HX motif. However, the authors argue that the HX motif has appeared late in the superfamily of Hox proteins, but this is difficult to reconcile with the evolution of these families since this would imply that the HX domain has evolved independently in multiple proteins, after the duplication of the precursor Hox*.

This comment can be addressed with the minor comments from Reviewer 2. He/she proposes the exact opposite: “In the context of the discussion, I wonder whether it really is so difficult to imagine that the hexapeptide evolved independently in distinct ANTP subclasses? It is, after all, a short sequence in which a high degree of variability can be seen amongst gene families and between taxa. In some instances it seems to effectively be just a single Tryptophan (W) residue. One scenario might be that once a hexapeptide–TALE interaction is established in one family then the hexapeptide-interacting aspect of the TALE protein is established and is retained whilst the hexapeptide evolves in other ANTP subclasses? This scenario is the compliment of the differential loss scenario described by the authors, in terms of explaining the patchy distribution of hexapeptide domains across the ANTP class.”

Here we presented two possible scenarios, taking into account the phylogeny history of ANTP class genes. In both models, we postulated that the HX was present in a common ancestor of Hox, NK and extended-Hox members. We did not argue that the HX appeared late in Hox proteins. We agree with Reviewer 2 that given the short size of the HX, this motif could have appeared independently in multiple times during evolution. However, we have to consider that this motif is always located at a precise position in all ANTP subclass members (upstream of the HD), which is probably important to ensure interactions with PBC members. Along the same line, the distance between the HX and the HD may also have played important roles in the acquisition of new functions, as noticed in certain bilaterian Hox proteins. For all these reasons, it is difficult to imagine that the invariant position of the HX could result from independent acquisitions during metazoan evolution. We tried to better emphasize this point in the corresponding discussion part.

*5) It seems that a bit more speculation would be warranted to place the work in context. For example, they do not touch the question of axis very much. In*
Figure 1
*they represent the aboral-oral axis of Cnidaria analogous to the anterior-posterior in bilateria. In the same figure they show evidence for asymmetry of Hox gene expression, along the directive axis, as it was also found by others. This would of course be in line with the enterocoel hypothesis and their short remark in the text would also support this. Hence, I would encourage them to add a paragraph on discussing axis homologies, although I do not expect that they* can *arrive at a final conclusion on this yet. Still, their work evidently contributes to this important question*.

We chose not to extrapolate too much with our results on the longstanding and heavy debate of the homologous comparison between bilaterian and cnidarian longitudinal axes. We have now discussed this point in more detail, providing references illustrating the controversy for both the primary and secondary axes of cnidarians. We concluded on the fact that patterning molecules (Hox, BMPs, etc) could simply be used along different longitudinal axes in different species. What is important is the existence of a patterning system that could be co-opted during evolution: “Together these observations highlight that patterning molecules can be used along different longitudinal axes during animal evolution, which renders difficult the comparison between Bilateria and Cnidaria. Here we propose that the ancestral molecular cues underlying the Hox patterning system along the cnidarian directive axis could have been recruited for AP patterning in Bilateria.”

*6) For the revisions, authors should revisit the question of which figures should be in the main text and which provide only supplementary information*.

We have changed the previous supplementary figures 9 and 11-13 into Figures 7 and 9, respectively. In addition, we have simplified the presentation of previous Figures 6 and 7, which are now regrouped in a new Figure 6, focusing on functional data in *Drosophila*. Preliminary in vivo and in vitro interaction assays are now included in figure supplements.